# Single-amino acid variants reveal evolutionary processes that shape the biogeography of a global SAR11 subclade

Tom O Delmont[1†], Evan Kiefl[1,2†], Ozsel Kilinc[3], Ozcan C Esen[1], Ismail Uysal[3], Michael S Rappé[4], Steven Giovannoni[5], A Murat Eren[1,6]*

[1]Department of Medicine, The University of Chicago, Chicago, United States; [2]Graduate Program in Biophysical Sciences, University of Chicago, Chicago, United States; [3]Department of Electrical Engineering, University of South Florida, Tampa, United States; [4]Hawaii Institute of Marine Biology, University of Hawaii at Manoa, Kaneohe, United States; [5]Department of Microbiology, Oregon State University, Corvallis, United States; [6]Marine Biological Laboratory, Woods Hole, United States

**Abstract** Members of the SAR11 order Pelagibacterales dominate the surface oceans. Their extensive diversity challenges emerging operational boundaries defined for microbial 'species' and complicates efforts of population genetics to study their evolution. Here, we employed single-amino acid variants (SAAVs) to investigate ecological and evolutionary forces that maintain the genomic heterogeneity within ubiquitous SAR11 populations we accessed through metagenomic read recruitment using a single isolate genome. Integrating amino acid and protein biochemistry with metagenomics revealed that systematic purifying selection against deleterious variants governs non-synonymous variation among very closely related populations of SAR11. SAAVs partitioned metagenomes into two main groups matching large-scale oceanic current temperatures, and six finer proteotypes that connect distant oceanic regions. These findings suggest that environmentally-mediated selection plays a critical role in the journey of cosmopolitan surface ocean microbial populations, and the idea 'everything is everywhere but the environment selects' has credence even at the finest resolutions.
DOI: https://doi.org/10.7554/eLife.46497.001

*For correspondence: meren@uchicago.edu

†These authors contributed equally to this work

**Competing interests:** The authors declare that no competing interests exist.

## Introduction

The SAR11 order *Pelagibacterales* (*Thrash et al., 2011*; *Ferla et al., 2013*) is one of the most ubiquitous free-living lineages of heterotrophic bacteria in the world's oceans (*Giovannoni et al., 1990*; *Morris et al., 2002*; *Carlson et al., 2009*; *Eiler et al., 2009*; *Schattenhofer et al., 2009*; *Treusch et al., 2009*). Successful cultivation efforts and single amplified genomes from the environment have led to studies revealing their critical role in marine carbon cycling (*Rappé et al., 2002*; *Giovannoni et al., 2005*; *Stingl et al., 2007*; *Oh et al., 2011*; *Tsementzi et al., 2016*; *White et al., 2019*), and environmental sequencing surveys have offered detailed insights into the ecology of this ancient branch of life in aquatic environments across the globe (*Zinger et al., 2011*; *Brown et al., 2012*).

The evolution of SAR11 is an active area of research (*Giovannoni, 2017*) that is critically important to understanding the determinants of its remarkable ability to maintain abundant populations in the global ocean. The evolutionary origins of SAR11 and thus its precise placement in the Tree of Life is debated (*Thrash et al., 2011*; *Rodríguez-Ezpeleta and Embley, 2012*; *Ferla et al., 2013*; *Viklund et al., 2013*), and our understanding of the evolutionary processes that define the biogeography of SAR11 cells is not complete. At the level of major SAR11 clades, previous studies have

attributed markedly distinct patterns of distribution in the global ocean to both niche-based (*Brown et al., 2012*; *Eren et al., 2013a*) and neutral processes (*Manrique and Jones, 2017*). At the level of individual populations, a key simulation by *Hellweger et al. (2014)* showed that the intra-population sequence divergence that reflects the geographic patterns of distribution for SAR11 cells could emerge solely as a function of ocean currents, without selection (*Hellweger et al., 2014*). Between the extremes of inter-clade and intra-population diversity lies a wealth of variation that potentially can yield insights into the ecological and genetic forces that determine genomic diversity and fitness between closely-related, naturally occurring SAR11 populations.

High-throughput sequencing of metagenomes provides access to genome-wide heterogeneity within environmental populations (*Simmons et al., 2008*), and current computational strategies can reveal associations between ecological parameters and microdiversity patterns at various levels of resolution (*Eren et al., 2015*; *Scholz et al., 2016*; *Nayfach et al., 2016*; *Costea et al., 2017*; *Truong et al., 2017*). However, SAR11 poses multiple challenges for such investigations, including their remarkable intra-population genomic diversity and the limited success of reconstructing SAR11 genomes from metagenomic data. Comprehensive investigations of the genetic contents of naturally occurring microbial populations (see *Denef, 2018*) for a review) often rely on population genomes directly reconstructed from metagenomes (*Simmons et al., 2008*; *Bendall et al., 2016*; *Anderson et al., 2017*; *Garcia et al., 2018*). While advances in genome-resolved metagenomics have made microbial clades more accessible without cultivation (*Spang et al., 2015*; *Brown et al., 2015*; *Anantharaman et al., 2016*), reconstructing SAR11 genomes from the surface ocean remains a difficult endeavor, as evident in recent comprehensive surveys of metagenome-assembled genomes (MAGs) from seawater samples from around the globe (*Tully et al., 2018*; *Delmont et al., 2018*). In the absence of population genomes recovered directly from the environment, genomes from isolates can also offer insights into environmental populations through genome-wide recruitment analyses in which short metagenomic reads are aligned to a reference (*Denef, 2018*).

Using metagenomic read recruitment to investigate the structure of environmental populations is confounded by the challenge of defining the boundaries of microbial populations. Without an established species concept in microbiology, defining units of microbial diversity and their boundaries is a significant challenge (see *Shapiro, 2018* and *Cohan, 2019* for discussions). Nevertheless, from analyses of isolated microbial strains with formal taxonomic descriptions, a genome-wide average nucleotide identity (gANI) cutoff of 95% emerged as an operational delineation of species (*Konstantinidis and Tiedje, 2005*; *Varghese et al., 2015*) and was confirmed in a recent analysis of eight billion pairwise comparisons of whole genomes (*Jain et al., 2018*). Both gANI calculations using complete genomes, as well as the average nucleotide identity of metagenomic short reads (ANIr) recruited from environmental metagenomes using reference genomes, show an interesting discontinuity among sequence-discrete populations at sequence identity levels between 80% and 90–95% (*Konstantinidis and DeLong, 2008*; *Caro-Quintero and Konstantinidis, 2012*; *Jain et al., 2018*). Regardless of their theoretical significance, these cutoffs are essential for multiple practical purposes, such as the identification and subsequent exclusion of metagenomic reads that originate from non-target environmental populations, to avoid inflating variants arising from contaminating non-specific reads in microbial population genetics studies.

Interestingly, the boundaries of environmental SAR11 populations appear to not comply with the 95% ANIr cutoff. For instance, *Tsementzi et al. (2016)* observed substantial sequence diversity within sequence-discrete SAR11 subclades in the environment, and suggested that an ANIr as low as 92% would be required to adequately define the boundaries of the SAR11 populations recovered in their study (*Tsementzi et al., 2016*). These findings are consistent with a comprehensive study of isolate genomes and marine metagenomes by *Nayfach et al. (2016)*, which suggested that SAR11 is one of the most genetically heterogeneous marine microbial clades (*Nayfach et al., 2016*). The substantial sequence diversity within environmental SAR11 populations not only explains the absence of SAR11 population genomes in genome-resolved metagenomics studies, but also challenges conventional approaches to the study of population genetics in microorganisms. For instance, the multiple occurrence of single-nucleotide variants in individual codon positions would render commonly used computational strategies that classify synonymous and non-synonymous variations based on independent nucleotide sites (such as in *Schloissnig et al., 2013*; *Bendall et al., 2016*) unfeasible. Despite these challenges, SAR11, with its ubiquity in surface seawater samples, extensive

diversity in sequence space, and unique evolutionary history, remains one of the exciting puzzles of contemporary microbiology.

Here we investigated the evolutionary processes that maintain genetic diversity within a natural SAR11 lineage accessible through a single isolate genome that recruited more than 1% of surface ocean metagenomic reads from a global dataset. Using single-amino acid variants, we were able to (1) delineate multiple proteotypes whose distributions were more closely linked to large-scale oceanic current temperatures than they were to geographic proximity, and (2) resolve positive and negative selection mediated by temperature and its co-variables. Our findings suggest that environmentally mediated selection, rather than neutral processes, dominate the biogeographic partitioning of SAR11 at fine scales of taxonomic resolution. Our study also offers new computational approaches to characterize variation within complex microbial populations, including additional means to integrate amino acid and protein biochemistry into microbial population genetics.

## Results and discussion

To find the most appropriate SAR11 isolate genome to study the population genetics of naturally occurring SAR11, we used the complete genomes of 21 SAR11 isolates in a competitive recruitment of short reads from 103 metagenomes. Most of these metagenomes were from the TARA Oceans Project (*Sunagawa et al., 2015*), and correspond to 93 stations across four oceans and two seas. We also included an additional 10 metagenomes from the Ocean Sampling Day Project (*Kopf et al., 2015*) to cover high-latitude areas of the Northern hemisphere. All metagenomes correspond to small planktonic cells (0.2–3 μm in size) from the surface (0–15 meters depth; n = 71) and deep chlorophyll maximum (17–95 meters depth; n = 32) layers of the water column (*Supplementary file 1a*). The isolates we used belonged to SAR11 subclades Ia.1 (n = 6), Ia.3 (n = 11), II (n = 1), IIIa (n = 2) and the related alphaproteobacterium Va (n = 1) (*Supplementary file 1b*), which collectively recruited 1,029,716,339 reads from all metagenomes, or 3.3% of the dataset (*Supplementary file 1c*).

### The metapangenome of SAR11

To investigate associations between ecology and gene content of SAR11 lineages, we first performed a pangenomic analysis in conjunction with read recruitment from the metagenomic data. The pangenome of SAR11 genomes consisted of all 29,719 genes grouped into 6175 gene clusters (*Supplementary file 1d*). The clustering of genomes based on shared gene clusters (*Supplementary file 1e*) matched that of the previously described phylogenetic clades (*Grote et al., 2012*) (*Figure 1A*; an interactive version of which is available at http://anvi-server.org/p/4Q2TNo). The SAR11 pangenome across metagenomes (i.e., the SAR11 metapangenome) revealed distinct distribution patterns for each clade within SAR11 (*Figure 1A*). Clade Ia recruited the most reads compared to other clades (*Supplementary file 1b*), consistent with previous studies that found this clade to be highly abundant in surface seawater (*Field et al., 1997*; *Brown et al., 2012*; *Eren et al., 2013a*; *Manrique and Jones, 2017*). Gene clusters divided clade Ia into two main clusters corresponding to the high-latitude subclade Ia.1 and the low-latitude subclade Ia.3 (*Figure 1A*). While all high-latitude genomes displayed a bi-polar geographic distribution in the metagenomic dataset, gene clusters in low-latitude genomes revealed multiple sub-groups that also showed different patterns of geographic distribution (*Figure 1A*). This emphasized the need to further refine subclade 1a.3, in which each genome pair had over 98.6% sequence identity at the 16S rRNA gene level (*Supplementary file 1f*). Our consideration of geographical co-occurrence patterns, phylogenomic characteristics, and pangenomic properties in this metapangenome revealed six subclades within 1a.3 with cultured representatives (*Figure 1A*, also see *Supplementary file 1g* for gANI estimates between SAR11 genomes). We tentatively name them SAR11 subclade 1a.3.I (HTCC7211, HTCC7214 and HTCC7217; gANI of >93% and 16S rRNA gene identity of >99.4%), 1a.3.II (HIMB5), 1a.3.III (HIMB4 and HIMB1321; gANI of 94.8% and 16S rRNA gene identity of 100%), 1a.3.IV (HTCC8051 and HTCC9022; gANI of 86.9% and 16S rRNA gene identity of 100%), 1a.3.V (HIMB83) and 1a.3.VI (HIMB122 and HIMB140; gANI of 94.6% and 16S rRNA gene identity of 99.7%). Overall, the refinement of SAR11 subclades reveals a striking agreement between phylogeny, pangenome, and the ecology of the members of the SAR11 clade Ia.

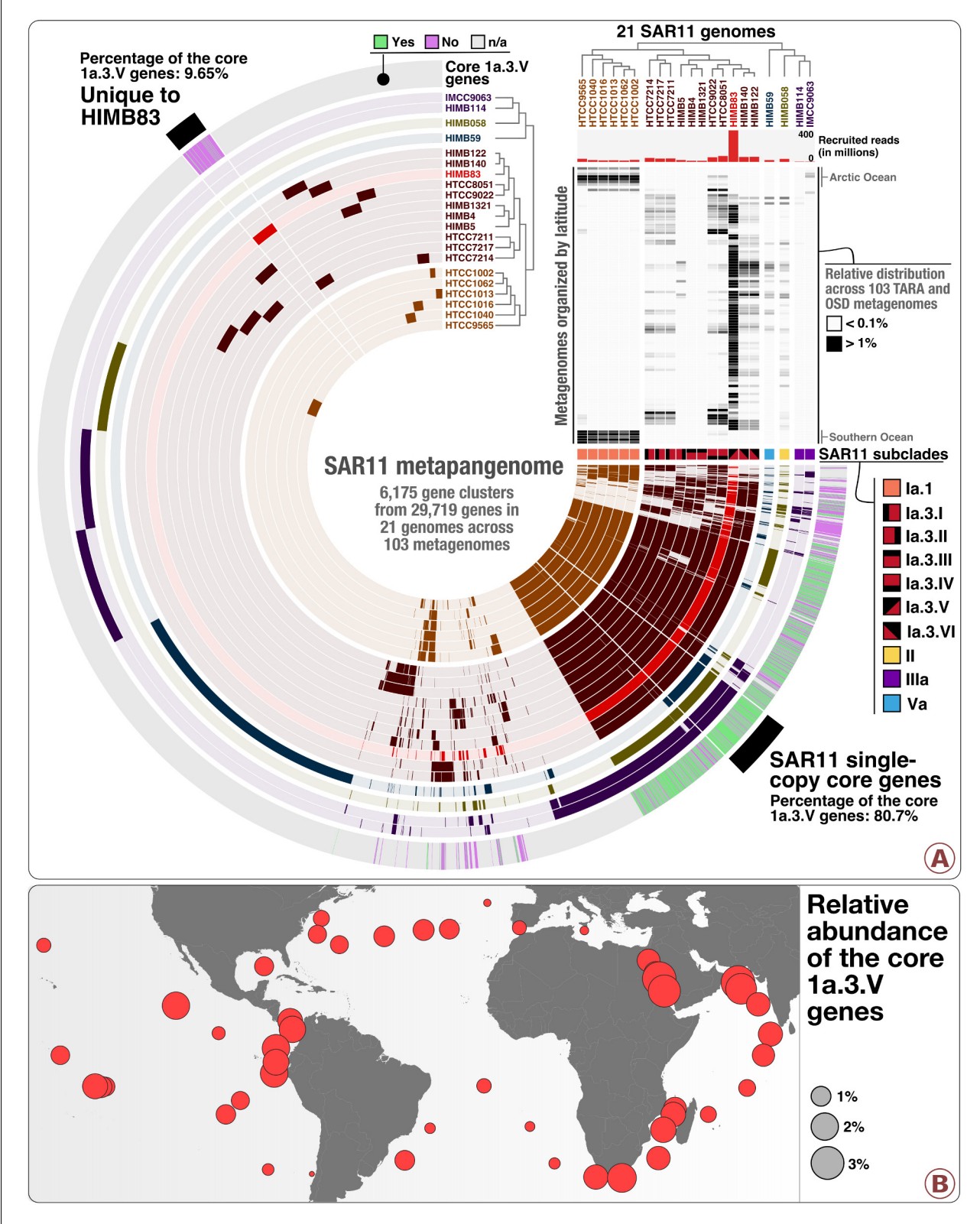

**Figure 1.** The SAR11 metapangenome. Panel A describes the pangenome of 21 SAR11 isolate genomes based on the occurrence of 6175 gene clusters, in conjunction with their phylogeny (clade level) and relative distribution of recruited reads in 103 metagenomes ordered by latitude from the North Pole to the South Pole (top right heat map). The relative distributions were displayed for a minimum value of 0.1% and a maximum value of 1%. The layer named 'Core 1a.3.V genes' displays the occurrence of the 799 core 1a.3.V genes (in green) and those found in HIMB83 but not in the 1a.3.V

*Figure 1 continued on next page*

*Figure 1 continued*
lineage (in purple). Panel B describes the relative distribution of reads the 799 core 1a.3.V genes recruited across surface metagenomes from TARA Oceans.
DOI: https://doi.org/10.7554/eLife.46497.002
The following figure supplement is available for figure 1:
**Figure supplement 1.** Distribution and diversity of the core 1a.3.
DOI: https://doi.org/10.7554/eLife.46497.003

## A remarkably abundant and widespread SAR11 lineage at low latitudes

While Ia.3 was the most abundant SAR11 subclade in our dataset, the new subclades we defined in this group differed remarkably in their competitive recruitment of short reads from metagenomes (*Figure 1A*, *Supplementary file 1b*). For example, while the least abundant subclade (1a.3.II; represented by HIMB5) recruited 22.6 million reads, the most abundant one (1a.3.V; represented by HIMB83), recruited 390.9 million reads, or 1.18% of the entire metagenomic dataset (*Supplementary file 1b*). For perspective, this is roughly two times more reads than the most abundant *Prochlorococcus* isolate genome recruited from the same dataset (*Delmont and Eren, 2018*) (*Supplementary file 1h*). Strain HIMB83 contains a 1.4 Mbp genome with 1470 genes, and was isolated from coastal seawaters off Hawai'i, USA. But it also recruited large numbers of reads from locations that were distant to the source of isolation (*Supplementary file 1c*). The gANI between HIMB83 and the most similar genome in our dataset, HIMB122 (1a.3.VI) was 82.6%, and the remarkable abundance of HIMB83 has also been recognized by others (*Brucks, 2014*; *Nayfach et al., 2016*). To the best of our knowledge, 1a.3.V is the most abundant and widespread SAR11 subclade in the euphotic zone of low-latitude oceans and seas.

Although it is a member of the subclade 1a.3.V, the genomic context HIMB83 provides does not exhaustively describe the gene content of all members of 1a.3.V. Nevertheless, it gives access to the core 1a.3.V genes through read recruitment. To identify core 1a.3.V genes, we used a conservative two-step filtering approach. First, we defined a subset of the 103 metagenomes within the main ecological niche of 1a.3.V using genomic mean coverage values (*Supplementary file 1c*). Our selection of 74 metagenomes in which the mean coverage of HIMB83 was >50X encompassed three oceans and two seas between −35.2° and +43.7° latitude, and water temperatures at the time of sampling between 14.1°C and 30.5°C (*Figure 1—figure supplement 1*, *Supplementary file 1i*). We then defined a subset of HIMB83 genes as the core 1a.3.V genes if they occurred in all 74 metagenomes and their mean coverage in each metagenome remained within a factor of 5 of the mean coverage of all HIMB83 genes in the same metagenome. This criterion accounted for biological characteristics influencing coverage values in metagenomic surveys of the surface ocean such as cell division rates and variations in coverage as a function of changes in GC-content throughout the genomic context. *Figure 1—figure supplement 1* displays the coverage of all HIMB83 genes across all metagenomes, and *Supplementary file 1j* reports underlying coverage statistics. While the 799 genes that met these criteria systematically occurred within the niche boundaries of 1a.3.V, 40% of the remaining 671 HIMB83 genes that were filtered out were present in five or fewer metagenomes and coincided with hypervariable genomic loci (*Figure 1—figure supplement 1*). Hypervariable genome regions are common features of surface ocean microbes (*Coleman et al., 2006*; *Zaremba-Niedzwiedzka et al., 2013*; *Kashtan et al., 2014*; *Delmont and Eren, 2018*) that are not readily addressed through metagenomic read recruitment but do influence pangenomic trends. Here, less than 10% of gene clusters unique to HIMB83 were among core 1a.3.V genes (*Figure 1A*), indicating HIMB83's unique genes are mostly accessory to the members of 1a.3.V. In contrast, more than 80% of gene clusters that were core to the 21 SAR11 genomes matched to the core 1a.3.V genes. The overlap between environmental core genes of 1a.3.V revealed by the metagenomic read recruitment and the genomic core of SAR11 revealed by the pangenomic analysis of isolate genomes suggests that these genes represent a large fraction of the 1a.3.V genomic backbone (*Figure 1A*). Core 1a.3.V genes recruited on average 1.25% of reads in the 74 metagenomes (*Figure 1B*, *Supplementary file 1j*). The broad geographic prevalence of core 1a.3.V genes represents a unique opportunity to study the population genetics of an abundant marine microbial subclade across distant geographies.

## SAR11 subclade 1a.3.V maintains a substantial amount of genomic heterogeneity

To investigate the amount of genomic heterogeneity within 1a.3.V, we first studied individual short reads that the HIMB83 genome recruited from metagenomes. The percent identity of reads that matched to the 799 core 1a.3.V genes ranged from 88% to 100% (*Figure 2*), which is considerably more diverse than those observed in similar reference-based metagenomic studies (*Konstantinidis and DeLong, 2008*; *Tsementzi et al., 2016*; *Meziti et al., 2019*). Notably, we also observed similar trends for the other SAR11 genomes included in this study (*Figure 2—figure supplement 1*), suggesting that the relatively high sequence diversity observed among core 1a.3.V genes may be a characteristic shared with other SAR11 lineages in the surface ocean.

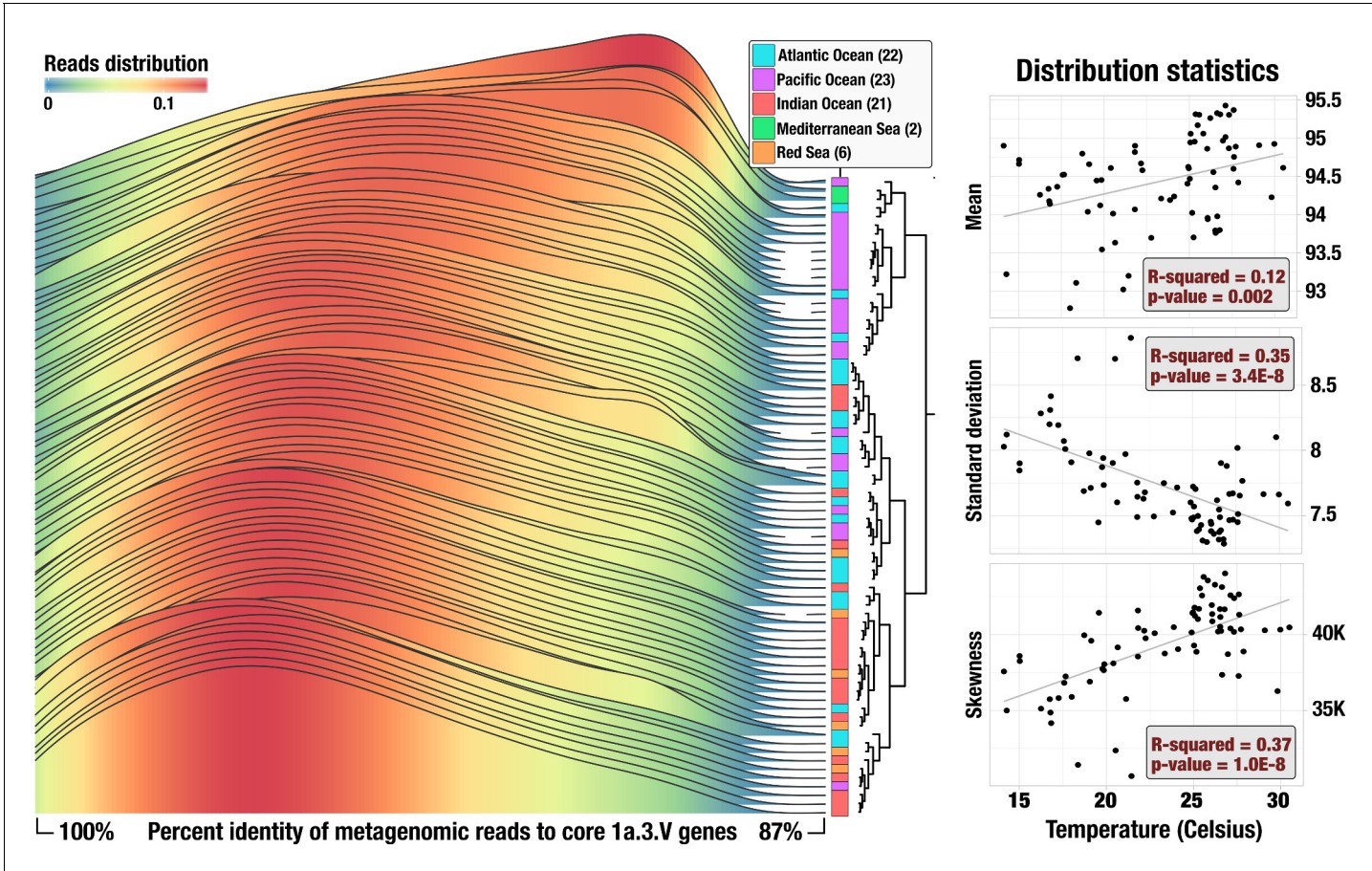

**Figure 2.** Statistics of recruited reads. Left panel shows percent identity distributions in each of the 74 metagenomes. Curves are colored based on height. Metagenomes are ordered according to how the percent identity distributions hierarchically cluster based on Euclidean distance (dendrogram). Right panels display a summary of distribution statistics for each percent identity distribution compared against in situ temperature in a linear regression (correlations to all other available parameters are summarized in *Figure 2—figure supplement 2*). Each point is a metagenome and black lines are lines of best fit. For visual clarity, the data in left panel considers only the median read length and interpolates between data points, whereas the data in right panels consider all read lengths with no interpolation.

DOI: https://doi.org/10.7554/eLife.46497.004

The following figure supplements are available for figure 2:

**Figure supplement 1.** Percent identity distributions resulting from the competitive mapping experiment of the metagenomic short reads onto the 21 SAR11 reference genomes.

DOI: https://doi.org/10.7554/eLife.46497.005

**Figure supplement 2.** A matrix illustrating the degree of correlation (via linear regression) between oceanic metadata and the statistics (mean, standard deviation, skewness) of the percent read identity distributions of reads recruited by HIMB83 for the 74 metagenomes in which HIMB83 was covered at least 50X.

DOI: https://doi.org/10.7554/eLife.46497.006

Overall, our data confirm that ANIr values of >95% used previously to delineate sequence-discrete populations does not apply to SAR11. One immediate implication of this substantial amount of sequence diversity that defies previous empirical observations is our inability to explicitly define what we are accessing in the environment. This challenge is partially because a precise and exhaustive description of what constitutes a 'population' remains elusive (*Cohan and Perry, 2007*; *Shapiro and Polz, 2015*; *Cohan, 2019*), which creates significant practical challenges (*Rocha, 2018*), such as the accurate determination of the boundaries of naturally occurring microbial populations especially in metagenomic read recruitment results. Nevertheless, the term 'population' is frequently used in literature (*Simmons et al., 2008*; *Kashtan et al., 2014*; *Bendall et al., 2016*), which implies that Charles Darwin's observation in his historical work '*On the Origin of Species*' continues to summarize our struggle in life sciences to describe theoretical boundaries of fundamental units of life even though contemporary enviornmental microbiology has gone beyond the term species in this pursuit: '*no one definition of species has yet satisfied all naturalists; yet every naturalist knows vaguely what [they mean] when [they speak] of a species*' (*Darwin, 1859*). Our study is not well-positioned to offer a precise theoretical definition for the term 'population', either. Instead, similar to previous studies, we resort to an operational definition that suggests a population is 'an agglomerate of naturally occurring microbial cells, genomes of which are similar enough to align to the same genomic reference with high sequence identity' (*Delmont and Eren, 2018*; also see *Denef, 2018* and references therein for a comprehensive discussion of what constitutes a population from a metagenomic perspective). By outsourcing the hypothetical radius of a population in sequence space to the minimum sequence identity of short reads recruited from metagenomes, this approach offers a practical means to study very closely related environmental sequences without invoking theoretical considerations. The broad heterogeneity continuum that possesses no discernible sequence-discrete components we observed within the narrow sequence set defined this way, i.e., the metagenomic reads that match competitively to conserved HIMB83 genes (*Figure 2*), supports the assumption that this set originates within a population boundary (*Figure 1—figure supplement 1*). However, due to the incomplete theoretical foundation and limitations associated with the use of short metagenomic reads, in discussions here we more conservatively assume that our reads originate from multiple closely related yet intertwined SAR11 populations within subclade 1a.3.V.

Both high recombination rates between cells displaying low gANI values and frequent transfer of adaptive genes between ecologically distinct clades could explain the high-level of cohesion between SAR11 populations in the surface ocean (*Cohan, 2019*; *Vergin et al., 2007*). The high density of closely related 1a.3.V cells in the surface ocean suggests the strength of these two forces could be high within populations as well. At least two hypotheses reconcile extensive SAR11 sequence diversity and aide in understanding its implications. One hypothesis is that the members of 1a.3.V we access are in the process of evolving into multiple sequence-discrete populations and we are simply observing an emerging fork in the evolutionary journey of SAR11. Alternatively, the observed diversity may represent a cloud of random sequence variants akin to a quasispecies (*Domingo et al., 2012*). To examine these hypotheses, we tested the correlation between basic statistical properties of these curves (i.e., mean, standard deviation, and skewness) and environmental parameters via linear regression (*Figure 2—figure supplement 2*, *Supplementary file 1k*). This analysis revealed a significant correlation between in situ temperature and distribution shape (mean p-value: $2.0 \times 10^{-3}$; standard deviation p-value: $3.4 \times 10^{-8}$; skewness p-value: $1.0 \times 10^{-8}$), which suggests a strong influence of temperature and its co-variables on the sequence heterogeneity within 1a.3.V (*Figure 2*) and is incompatible with the hypothesis of random sequence variants.

## SAAVs: Accurate characterization of non-synonymous variation

Percent identity distributions are useful to assess overall alignment statistics of short reads to a reference; however, they do not convey information regarding allele frequencies, their functional significance, or association with biogeography. To bridge this gap, we implemented a framework to characterize amino acid substitutions in metagenomic data and to study genomic variation that impacts amino acid sequences (see Materials and methods). Briefly, our approach employs only metagenomic short reads that cover all three nucleotides in a given codon to determine the frequency of single-amino acid variants (SAAVs) in translated protein sequences. While synonymity is a codon characteristic, in practice it is often determined from a single-nucleotide variant (SNV) with the assumption that the two remaining nucleotides are invariant. However, populations with

extensive nucleotide variation can violate this assumption. Indeed, in the case of the core 1a.3.V genes, on average 22.5% of SNVs per metagenome co-occurred with other SNVs in the same codon. Thus, quantifying frequencies of full codon sequences as implemented in the SAAV workflow is a requirement to correctly assess synonymy.

Among the 799 core 1a.3.V genes and 74 metagenomes, we identified 1,074,096 SAAVs in which >10% of amino acids diverged from the consensus (i.e., the most frequent amino acid for a given codon position and metagenome). The SAAV density (the percentage of codon positions that harbor a SAAV) of core 1a.3.V genes averaged 5.76% and correlated with SNV density (19.3% on average) across the 74 metagenomes (linear regression, p-value $<2.2 \times 10^{16}$; $R^2$: 0.90; *Figure 1—figure supplement 1* and *Supplementary file 1L*). SNV and SAAV density metrics did not decrease in metagenomes sampled closest to the source of isolation (*Supplementary file 2a, 2b, and 2c*), suggesting that the location of isolation for strain HIMB83 does not predict the biogeography and population genetics of 1a.3.V. To improve downstream beta-diversity analyses, we discarded codon positions if their coverage in any of the 74 metagenomes was <20X, which resulted in a final collection of 738,324 SAAVs occurred in 37,416 codon positions that harbored a SAAV in at least one metagenome among the total of 252,333 codon positions (14.8%) within the core 1a.3.V genes (*Supplementary file 2d*). We considered a protein to be 'invariant' (i.e., absence of variation due to intensive purifying or positive selection) in a given metagenome if it lacked SAAVs. They were rare in our data: in total, we detected 2,548 invariant proteins (only 4.3% of all possibilities across the 74 metagenomes) that encompassed only 113 genes (*Supplementary file 2e*). In addition, all genes, except one 679 nucleotide long ABC transporter (gene id 1469), contained at least one SAAV in at least one metagenome (*Supplementary file 2d*), revealing a wide range of amino acid sequence diversification among core 1a.3.V proteins.

## Hydrophobicity influences the strength of purifying selection acting on amino acids

To understand how commonly each amino acid was found in variant sites, we compared the amino acid composition of SAAVs to the amino acid composition of the core 1a.3.V genes (see Materials and methods). In a scenario in which amino acids are as common in SAAVs as they are across all 799 core genes, the frequency that an amino acid occurred in SAAVs (variant sites) would share one-to-one correspondence with its frequency within the core genes (all sites). While these variables were correlated (linear regression, p-value: $9.8 \times 10^{-6}$; $R^2$: 0.65), we observed large deviations from this null expectation, implying strong differential occurrence of amino acids in SAAVs relative to their occurrence in core genes (*Figure 3A*, *Figure 3—figure supplement 1*, *Supplementary file 2f*). All negatively charged (Asp, Glu) and uncharged polar (Thr, Asn, Ser, Gln) amino acids were significantly enriched in SAAVs compared to the core 1a.3.V genes (*Figure 3A*). For instance, while asparagine made up only 6.34% of all amino acids in the core genes, on average 10.7% (±0.16%) of SAAVs involved asparagine substitutions across the 74 metagenomes (*Supplementary file 2f*). Interestingly, unlike negatively charged amino acids, positively charged amino acids did not exhibit substantial differences (<4% deviation between core 1a.3.V genes and SAAVs). Thus, hydrophilic amino acids were either overrepresented or exhibited little change in SAAVs with respect to their frequency within core genes. In stark contrast, all hydrophobic amino acids, with the very notable exceptions of isoleucine and valine, were underrepresented in SAAVs (*Figure 3A*, *Figure 3—figure supplement 1*, *Supplementary file 2f*).

Hydrophobic interactions within the solvent inaccessible core of proteins are known to be critical for maintaining the stability required for folding and activity, which enforces a strong purifying selection placed on mutations occurring in buried (solvent inaccessible) positions (*Bustamante et al., 2000*; *Chen and Zhou, 2005*; *Worth et al., 2009*). Since hydrophobic amino acids form the majority of buried positions, they are on average under stronger purifying selection, which is the likely explanation for the underrepresentation of hydrophobic amino acids within SAAVs. On the other hand, mutations in exposed (solvent accessible) positions on the surface of proteins are tolerated more, as they are less likely to disrupt protein architecture. Overall, our compositional analysis revealed that the occurrence of amino acids in SAAVs is roughly correlate with the occurrence of amino acids within the core 1a.3.V genes, and that deviations from this expectation are driven in part by levels of

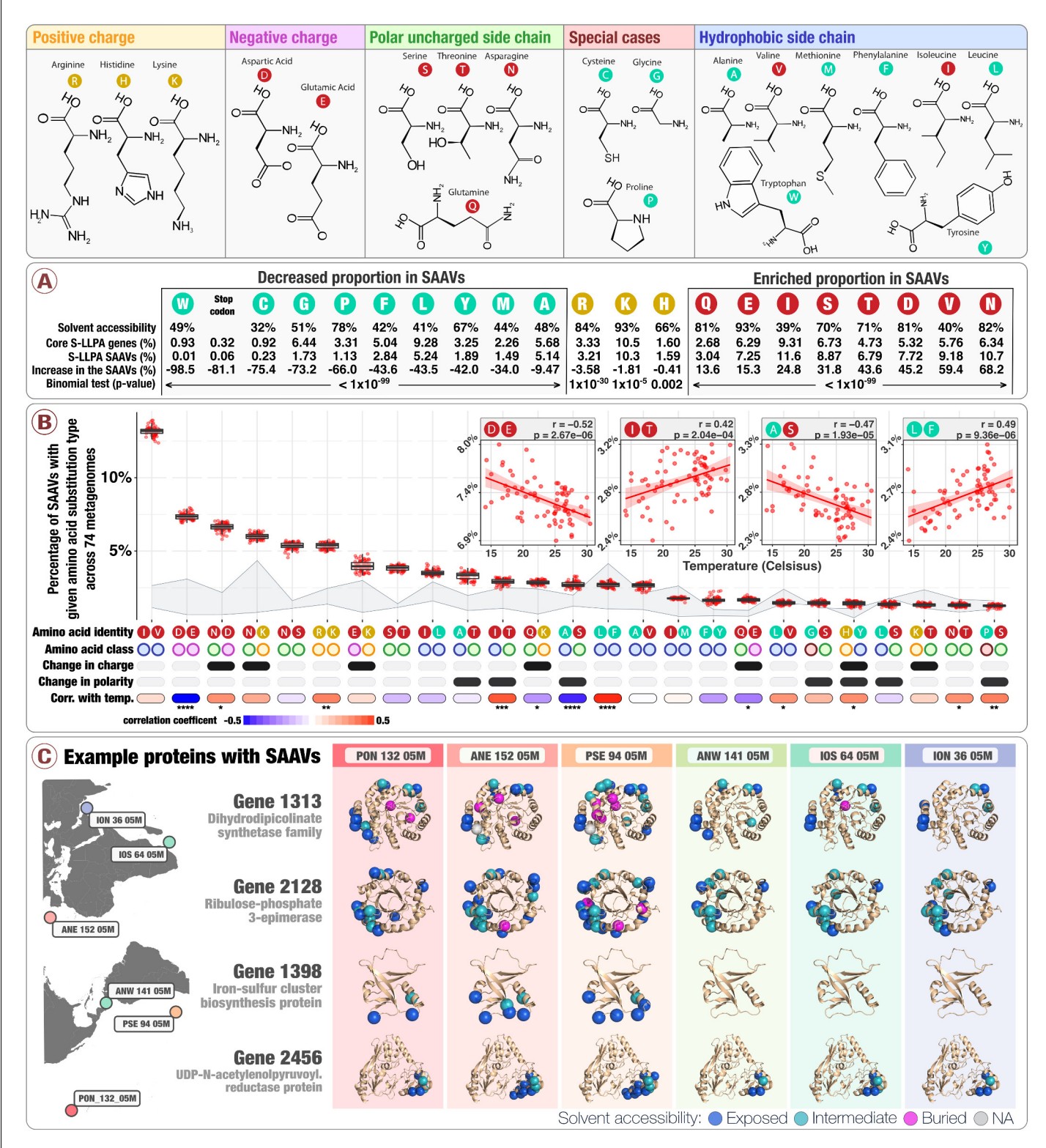

**Figure 3.** Physico-chemical properties of amino acid variants. The top panel describes the structure of 20 amino acids grouped by their main chemical properties. Panel A describes the solvent accessibility of amino acids, their relative distribution in both the core 1a.3.V genes and SAAVs, and their percentage increase in SAAVs as compared to the core 1a.3.V genes. The solvent accessibility of amino acids derives from the analysis of 55 proteins (**Bordo and Argos, 1991**). Panel B describes the relative abundance of the top 25 most prevalent amino acid substitution types (AASTs) across 74 metagenomes (boxplots), along with the classes their amino acids belong to and the correlation coefficient between AAST prevalence and in situ

*Figure 3 continued on next page*

*Figure 3 continued*

temperature calculated via linear regression (see *Figure 3—figure supplement 2* for p-values). The area shaded in light gray shows bounds for the expected frequency distribution given strictly neutral processes. The upper bound is Model one and the lower bound is Model 2 (see Materials and methods). The four insets example the relationship between AAST prevalence and in situ temperature for the AASTs 'aspartic/glutamic acid', 'isoleucine/threonine', 'alanine/serine', and 'leucine/phenylalanine' (*Figure 3—figure supplement 2* illustrate similar plots for all 25 of the most prevalent AASTs). The 25 AASTs included in the analysis cover 87.1% of all SAAVs. Panel C displays SAAVs on the predicted protein structures of four core 1a.3.V genes across six metagenomes from distant locations.

DOI: https://doi.org/10.7554/eLife.46497.007

The following figure supplements are available for figure 3:

**Figure supplement 1.** Panel A shows a direct comparison between the amino acid composition in all positions compared to the amino acid composition within SAAVs.

DOI: https://doi.org/10.7554/eLife.46497.008

**Figure supplement 2.** The top 25 most abundant amino acid substitution types (AASTs) and their relationship with in situ temperature.

DOI: https://doi.org/10.7554/eLife.46497.009

**Figure supplement 3.** Allele frequency trajectories and in situ temperature.

DOI: https://doi.org/10.7554/eLife.46497.010

**Figure supplement 4.** Analysis of how temperature-correlated variant positions distribute within Gene 1727, a glycine betaine ATP-binding cassette permease subunit identified for its rare proportion of temperature-correlated variant positions.

DOI: https://doi.org/10.7554/eLife.46497.011

purifying selection that depend upon the suitability of an amino acid's hydrophobicity for a given physicochemical environment (*Figure 3—figure supplement 1*).

## Amino acid exchange rates reveal hallmarks of neutral, purifying, and adaptive evolution

Next, we sought to investigate amino acids that co-occur in variable sites. SAAVs were often dominated by a few amino acids; hence, the frequency vector for a given SAAV contained many zero values. To reduce sparsity, we first simplified our data by associating each SAAV with an amino acid substitution type (AAST), defined as the two most frequent amino acids in a given SAAV. In 738,324 SAAVs, we observed 182 of 210 theoretically possible unique AASTs and a highly skewed AAST frequency distribution (*Supplementary file 2g*, *Figure 3B* boxplots). For example, the two most frequent AASTs, 'isoleucine/valine' and 'aspartic/glutamic acid', together comprised 20% of all SAAVs (*Figure 3B*). This is not surprising, since the amino acids in both of these AASTs (1) are common in the genome, (2) share very similar chemical structure (both differing by only a single methylene bridge), and (3) can be substituted through a single nucleotide substitution. On the other hand, the 'glycine/tryptophan' pair represents an opposite example: these amino acids (1) are uncommon in the genome, (2) share no chemical or structural similarity to one another, and (3) can only be substituted through a triple nucleotide substitution. Expectedly, 'glycine/tryptophan' was exceedingly rare in our data and occurred only once in 738,324 SAAVs (*Supplementary file 2h*).

While such a skewed AAST frequency distribution cannot be explained by strictly random mutational process (*Figure 3B* light-gray shaded area), it is compatible with standard theories of neutral or nearly-neutral evolution, since such theories consider the role of purifying selection (*Ohta and Gillespie, 1996*). Within subclade 1a.3.V the distribution of AAST frequencies was notably constrained across geographies (*Figure 3B*). For example, the relative standard deviation of 'aspartic/glutamic acid' frequencies across the 74 metagenomes was just 3.0%, and the statistical spread of other AASTs was comparable (*Figure 3B*). The overall consistency of AAST frequency distributions across geographies supports the hypothesis that purifying selection controls the permissibility of amino acid exchangeability within 1a.3.V and enables an interpretation of these data through a neutral model: SAAVs composing the AAST frequency distribution represent primarily neutral mutations that have drifted to measurable levels, and the lack of SAAVs in AASTs of dissimilar amino acids that likely represent deleterious mutations reflect the influence of purifying selection. However, a closer inspection reveals a subtle divergence of amino acid exchangeabilities that correlates with water temperature and/or its co-variables (*Figure 3B* insets, *Figure 3—figure supplement 2*). Note that this divergence is AAST specific; for example, positions with mixed proportions of glutamic and aspartic acid are less commonly found in warm waters (linear regression, uncorrected

p-value: $2.7 \times 10^{-6}$), yet for isoleucine and valine such a correlation is nonexistent (linear regression, uncorrected p-value: 0.418). These findings suggest that amidst a signal that is predominantly indicative of purifying selection, there appears to be a fingerprint of adaptive/divergent processes caused by temperature and/or its co-variables that subtly shift the mutational profile within 1a.3.V. We were unable to attribute the magnitude or direction of these correlations to differences between amino acids (i.e., changes in hydrophobicity, size, or charge). This was likely due to the insufficiency of characterizing SAAVs with only the chemical properties of the involved amino acids, and disregarding position-specific information, such as the surrounding physicochemical environment that can only be studied with knowledge of the protein's structure.

To address this shortcoming, we next sought to link SAAVs to predicted protein structures of the core 1a.3.V genes, 436 of which had significant matches in Protein Data Bank for template-based structure modeling (see Materials and methods). Placing SAAVs on predicted protein structures revealed that their occurrence was not randomly distributed but was instead strongly dependent on the local physicochemical environment of the structure (*Figure 3C*, *Supplementary file 3a* and http://data.merenlab.org/sar11-saavs). Within the subset of the 1a.3.V proteome accessible to us, we found that buried amino acids (0-10% relative solvent accessibility) were approximately 4.4 times less likely to be variant than those that were exposed (41-100% relative solvent accessibility) (ANOVA, p-value: $<2 \times 10^{-16}$). This observation was strikingly apparent in TIM barrels, where SAAVs mostly occurred in the outer alpha helix and loop regions (e.g., *Figure 3C* gene 2,128). This trend directly confirmed our previous inference (based on the underrepresentation of hydrophobic amino acids) that solvent inaccessible positions are subject to higher levels of purifying selection and thus contain fewer SAAVs. The local physicochemical environment therefore shapes variation, and visual inspection of *Figure 3C* indicates that this is conserved across distant geographies; that is positions that vary in one metagenome are likely to vary in others, as well. Overall, 91.7% of variant positions in the core 1a.3.V genes varied in 10 or more metagenomes, and 21.7% varied in all 74 metagenomes (*Supplementary file 2i*).

## Temperature correlates with amino acid allele frequency trajectories

In addition to considering patterns of variability that emerged when we pooled data across 37,416 codon positions exhibiting variation within the core 1a.3.V genes, we also investigated the allele frequency trajectories of individual positions (i.e., the relative frequency between the two most prevalent amino acids across the 74 metagenomes) and sought to identify those that correlate with in situ temperature and/or its co-variables. Amino acid allele frequencies in 4592 of the 37,416 positions were correlated with temperature (*Supplementary file 3b*; Benjamini–Hochberg multiple testing correction on linear regression p-values, false discovery rate 5%). *Figure 3—figure supplement 3* illustrates example cases and correlation statistics per AAST. It is statistically implausible that such correlations with temperature could have arisen from neutral evolution, given that distant oceans share similar temperatures (*Supplementary file 1a*). It is therefore most plausible to conclude that these allele frequency trajectories are the result of environmentally mediated selection. Although we note that, considering the pervasive effect of genetic hitchhiking in microbial evolution (*Good et al., 2017*), variation in a considerable fraction of positions may be neutral despite their association with temperature.

We then sought to investigate which positions are under selection, and whether the variation at these positions can be explained by differing levels of purifying selection, or diversifying selection that could be evidence of adaptive evolution. Scrutinizing all 4592 positions to address these critical questions is an intractable problem, so we narrowed our focus to genes possessing disproportionately high ratios of temperature-correlated to temperature-uncorrelated SAAV positions, since we expected this to be a reasonable criterion for identifying likely candidates of adaptive evolution (*Supplementary file 3c*). Of the 10 genes fitting this criterion (see Materials and methods), the permease subunit of a glycine betaine ATP-binding cassette (ABC) transporter stood out due to its appreciated relevance to SAR11 biology: glycine betaine transporters of SAR11 are highly translated proteins in the environment and transport osmolyte compounds into cells for energy production (*Noell and Giovannoni, 2019*). To investigate the positioning of amino acids in the tertiary structure of the permease relative to the cellular membrane, we first categorized the location of each residue as transmembrane, cytosolic (inside the inner membrane), or periplasmic (outside the inner

membrane) (*Figure 3—figure supplement 4*). Positions that were not correlated with temperature were commonly transmembrane, and infrequently periplasmic. In contrast, most positions that correlated with temperature were periplasmic (*Figure 3—figure supplement 4*). The probability of observing a similar distribution between temperature-correlated and temperature-uncorrelated positions across transmembrane, periplasmic, and cytosolic regions was only 0.034 (analytic trinomial test, temperature-uncorrelated distribution as prior), which indicates temperature-correlated positions are subjected to unique evolutionary forces. A previous study suggested that periplasmic residues of transmembrane proteins undergo higher rates of adaptive evolution due to their increased exposure to changing environmental conditions (*Sojo et al., 2016*). This observation lends additional support to the hypothesis that periplasmic SAAV positions within this gene that correlate with temperature are more likely shaped by adaptive processes.

Allele frequency trajectories also provide an opportunity to study the directionality of exchange rates of AASTs. For example, of the 1066 positions dominated by 'alanine/serine' SAAVs, 158 positions correlated with temperature (*Figure 3—figure supplement 3*). If there was no temperature-driven preference for either amino acid in this subset of positions, the frequency of alanine should positively correlate with temperature as often as the frequency of serine does. Yet this expectation is grossly violated: in 103 of 158 positions alanine frequencies positively correlated with temperature (binomial test, Bonferroni-corrected p-value: 0.004). Overall, this result indicates temperature-dependent amino acid substitution preferences that are independent of site (*Figure 3—figure supplement 3*).

## SAAV partitioning between warm and cold currents

We finally sought to extend the concept of allele frequency tracking at individual SAAV positions to investigate large-scale geographic partitioning of metagenomes. For this, we simplified the 738,324 SAAVs into a presence-absence matrix for codon position-specific AASTs across 74 metagenomes (*Supplementary file 2i,* also see 'Recovering codon position-specific AASTs from SAAVs' in Materials and methods). Of 57,277 codon position-specific AASTs affiliated with 37,415 unique codon positions, we detected 1.94% in all 74 metagenomes, while 33.3% were found in single metagenomes (*Supplementary file 2i*). To estimate distances between metagenomes based on these data, we used a Deep Learning approach. Briefly, this approach relies on a graph-based activity regularization technique for competitive learning from hyper-dimensional data, modified to reveal latent groups of variants in a fully unsupervised manner through frequent random sampling of variants (*Kilinc and Uysal, 2017*). Hierarchical clustering of samples based on Deep Learning-estimated distances (*Supplementary file 4a)* resulted in two main groups: the Western (warm) and Eastern (cold) boundary currents (*Figure 4A*). High latitude, relatively cold, and relatively nutrient rich waters are the source of Eastern boundary currents, which warm up and typically decline in nutrients as they transit in an equatorial direction. The opposite is true of Western boundary currents, which move poleward. The first group of 41 metagenomes, which matched cold currents (Benguela, Canary, California and Peru), encompassed most metagenomes from the Eastern Pacific Ocean, as well as the East side of the Atlantic Ocean (except near the southern tip of Africa) and the Mediterranean Sea (*Figure 4B*). The second group of 33 metagenomes, which matched warm currents (Agulhas, Somali, Mozambique, Brazil and Gulf stream), encompassed all metagenomes from the Red Sea and Indian Ocean, as well as metagenomes from the West side of the Atlantic Ocean (*Figure 4B*). Samples collected from the deep chlorophyll maximum layer of the water column mirrored trends observed in the surface samples (*Figure 4—figure supplement 1*). The association between SAAVs and ocean current type revealed a strong, global signal at the amino acid-level for 1a.3.V and suggested the presence of two main ecological niches for this lineage. Warm and cold currents are dynamic environments that differ in a host of factors in addition to the latitude and temperature of source waters. Factors that could drive adaptive changes in amino acid sequences between warm and cold currents include major differences in phytoplankton communities, altered composition of dissolved organic carbon pools, and the water temperature itself. Interestingly, the niche defined by cold currents exhibited significantly more SAAVs (ANOVA, p-value:$1.66 \times 10^{-12}$). This observation could be explained either by (1) extinction/re-emergence events that operate continually on specific codon positions (adaptive evolution), or (2) changes in abundances within a large seed bank of variants due to positive and negative selection as the lineage transits. A recent study using Lagrangian particle tracking and network theory suggested that all regions of the surface ocean are connected to each other with less

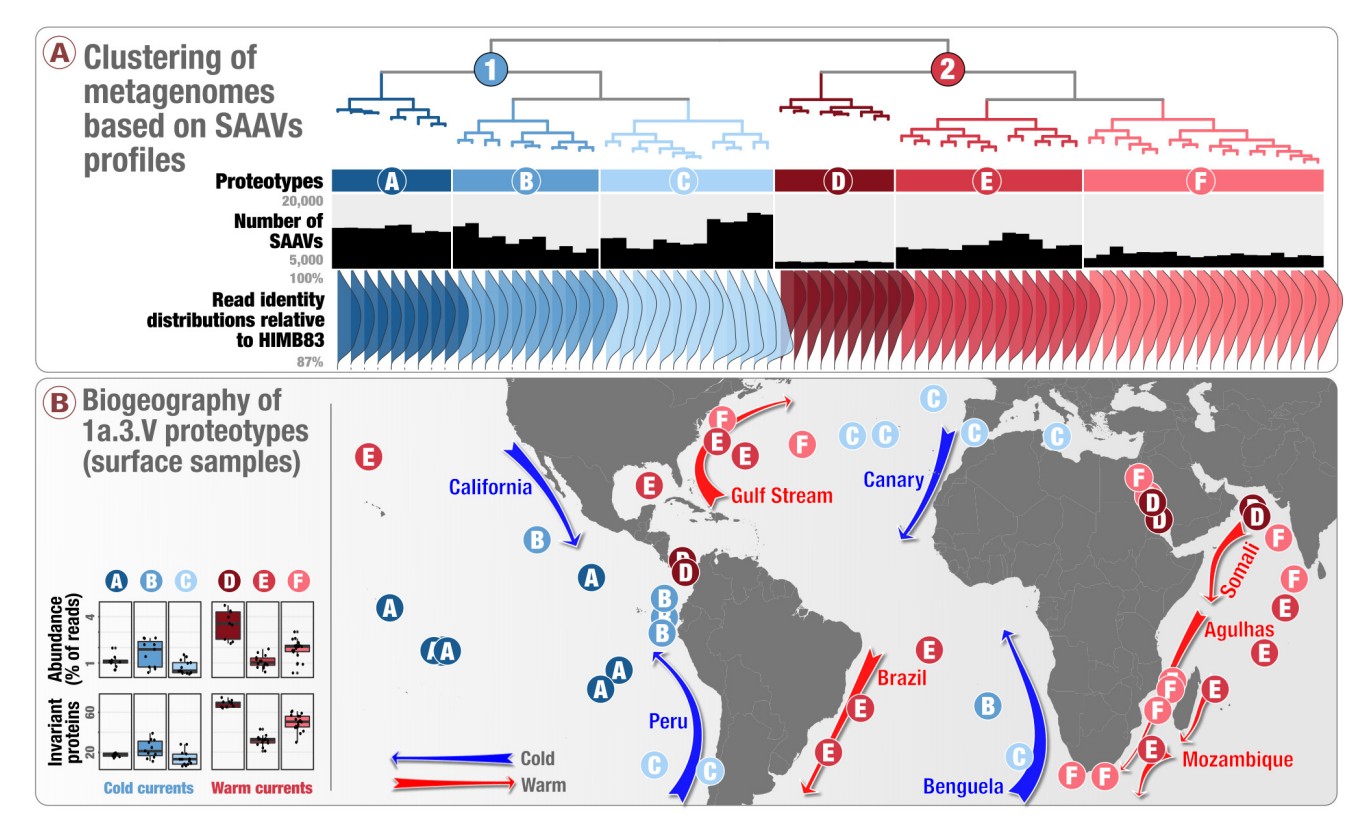

**Figure 4.** Biogeography of SAR11 subclade 1a.3.V based on single amino acid variants. Panel A describes the organization of 74 metagenomes based on 57,277 codon position-specific AASTs affiliated with 37,415 unique codon positions and summarizes the number of detected SAAVs and percent identity of reads HIMB83 recruited for each metagenome. The world map in panel B displays the geographic partitioning of the two main metagenomic groups and six proteotypes. Panel B also describes the relative abundance of 1a.3.V and the number of invariant proteins across the six proteotypes.

DOI: https://doi.org/10.7554/eLife.46497.012

The following figure supplements are available for figure 4:

**Figure supplement 1.** A comparison of the geographic partitioning of the 1a.3.

DOI: https://doi.org/10.7554/eLife.46497.013

**Figure supplement 2.** K-means clustering results (250 iterations) of the Deep Learning distance metric of 74 metagenomes based on the coordinates and identity of 738,324 SAAVs.

DOI: https://doi.org/10.7554/eLife.46497.014

**Figure supplement 3.** A comparison of dendrograms that organize metagenomes based on the genomic variability observed in the core 1a.3.

DOI: https://doi.org/10.7554/eLife.46497.015

**Figure supplement 4.** Biogeography of SAR11 subclade 1a.3.

DOI: https://doi.org/10.7554/eLife.46497.016

**Figure supplement 5.** Geographic partitioning of SAR11 by matching surface metagenomes analyzed in our study to simulated results determined using a neutral-agent based model (*Hellweger et al., 2014*).

DOI: https://doi.org/10.7554/eLife.46497.017

than a decade of transit (*Jönsson and Watson, 2016*), which might favor the latter scenario due to lack of time for the extinction and reemergence of variants in abundant marine microbial lineages.

To explore more detailed trends of the relationships between metagenomes, we further divided our dendrogram into six sub-clusters based on the elbow of the intra-cluster sum-of-squares curve of k-means clusters (*Figure 4—figure supplement 2*). These 1a.3.V 'proteotypes' grouped samples with similar amino acid variations (*Figure 4A*) and could not have been predicted from the clustering of samples based on percent identity distributions of short reads alone *Figure 4—figure supplement 3*). Among the environmental measurements for each metagenome (*Supplementary file 4b,*

*and 4c*), latitude and temperature at the time of sampling were the most significant predictors of the proteotypes (ANOVA, p-values: $8.56 \times 10^{13}$ and $3.57 \times 10^{-7}$, respectively).These two variables were followed by the concentrations of nitrate, phosphate, oxygen, and to a lesser extent, silicate and latitude (*Supplementary file 4d*). The number of SAAVs and the number of invariant proteins, however, were more significant predictors of these groups compared to all environmental parameters (ANOVA, p-values: $<2 \times 10^{-16}$, *Supplementary file 4c, and 4d*). Strikingly, most 1a.3.V proteotypes linked samples from distant geographical regions (*Figure 4B*). An exception to this was the proteotype A, which only contained Pacific Ocean metagenomes (*Figure 4B*). For instance, proteotypes E and F occurred both in the Indian Ocean and the West side of the Atlantic Ocean and associated with distinct warm currents: E was characteristic of the Mozambique and Brazil currents while F dominated the Agulhas current (*Figure 4B*). One of the most interesting proteotypes, D, whose reads most closely resembled the HIMB83 genome itself (*Figure 4B*), contained a distinctively low number of SAAVs, and grouped metagenomes sampled from both sides of the Panama Canal with metagenomes from the Red Sea and North of the Indian Ocean (*Figure 4B*). We also clustered the same data set using fixation index, a widely-used metric to measure population structure (*Weir, 2012*), which we modified in accordance with (*Schloissnig et al., 2013*) to permit multi-allelic variant positions. Both approaches preserved assocaitions between distant geographies (i.e., Proteotype D; *Figure 4* and *Figure 4—figure supplement 4*), however, they were not identical in their organization of metagenomes (i.e., Proteotype E was associated with colder currents according to fixation index rather than warmer ones; *Figure 4—figure supplement 4*), highlighting the non-trivial nature of establishing individual proteotypes from SAAVs.That said, the significance of in situ temperature to explain clustering of metagenomes into two main groups and six proteotypes was higher with Deep Learning (*Figure 4—figure supplement 4*), suggesting that Deep Learning was able to better capture the strong association between temperature and the genomic heterogeneity within 1a.3.V through SAAVs.

The striking connection between geographically distant regions of the oceans through SAAVs suggests a likely role for adaptive processes to maintain the genomic heterogeneity of closely related SAR11 populations within 1a.3.V (*Figure 4—figure supplement 5*). In fact, both the main ecological niches and more refined proteotypes indicate that SAAVs are not primarily structured by the global dispersal of water masses but instead tend to link distant geographic regions with similar environmental conditions (*Figure 4B*). Overall, these results indicate that environmentally-mediated selection is a strong determinant of SAR11 evolution and biogeography.

One question remains: what is the proportion of distinct evolutionary processes acting upon closely related SAR11 populations within 1a.3.V? Offering a precise answer to this critical question is compounded by multiple theoretical and technical factors. These factors include, but are not limited to, (1) the phenomenon of genetic hitchhiking that prevents accurate determination of amino acid positions that likely confer fitness, (2) the metagenomic short-read recruitment strategy that prevents absolute confidence regarding the origin of each fragment, (3) heavy reliance on temperature as the sole environmental stressor to predict associations between environmental parameters and variation due to limited insights into in situ physiochemistry, (4) the lack of a complete understanding of syntrophic relationships between taxa in the environment, and (5) computational bottlenecks to gain rapid and accurate insights into the role of variable amino acid residues even when protein structures are available. With these significant limitations in mind, we could nevertheless speculate that among the 252,333 total codon positions, 37,416 were variable, suggesting purifying selection maintains the conservancy of 85% of the positions within 799 core 1a.3.V genes. Of those 37,416 positions that were within the scope of permissible mutations, 4592 had amino acid frequency trajectories that significantly correlated with temperature, suggesting an upper-bound of 12% for the variable positions that are likely under the influence of temperature-driven adaptive processes, while neutral processes explain at least 88% of the variation. In summary, this global view of the data suggests that among the remarkable amount of variation within some of the most abundant and prevalent microbial populations in the ocean, adaptive evolutionary processes operating on core genes are responsible for variation in about 2% of all codon positions.

## Conclusions

We took advantage of billions of metagenomic reads to investigate single-amino acid variants (SAAVs) within the environmental core genes of the remarkably abundant and closely related SAR11

populations within subclade Ia.3.V, which we defined from a SAR11 metapangenome. The results elicit a highly-resolved quantitative description of purifying selection constraining the scope of permissible mutations to those that are not detrimental to protein stability requirements. Of permissible variation, thousands of codon positions harbored allele frequencies that systematically correlated with in situ temperatures and, overall, patterns of amino acid diversity reflected the temperature trends of large-scale ocean currents. This was especially apparent regarding the clear SAAV partitioning between Western and Eastern boundary currents. Previous studies have subdivided SAR11 clade Ia into cold-water (Ia.1) and warm-water (Ia.3) subclades with distinct latitudinal distributions (*Brown et al., 2012*), and reported sinusoidal oscillations between their abundances as a function of seawater temperature at a single temperate ocean site (*Eren et al., 2015*). At a much finer evolutionary scale (i.e., closely related populations within Ia.3.V), we observed significantly more protein variants in cold currents and more invariant proteins in warm currents, revealing a global pattern of alternating diversity for SAR11 in surface ocean currents in temperate and tropical latitudes. We were able to track this variation to changes in amino acid sequence preserved by selection.

Trends that emerged from our culture-independent survey of SAR11 were consistent with a recent study that also suggested an important role for environmental and ecological selective processes defining the spatial and temporal distribution of a widespread diatom species (*Whittaker and Rynearson, 2017*). Overall, these findings suggest that environmentally-mediated selection plays a critical role in the journey of cosmopolitan microbial populations in the surface ocean, lending credence to the idea for marine systems that 'everything is everywhere but the environment selects' (*Baas-Becking, 1934*). However, identifying environmental variables and their contributions to genomic heterogeneity within microbial populations is shrouded by both the dynamism and complexity of natural habitats, as well as the rich evolutionary dynamics that arise even in the simplest of conceivable environments (*Good et al., 2017*). These formidable challenges stress the importance of designing appropriate experiments to uncover variables that underpin the evolutionary divergence of closely related lineages, and drive transitions between them through space and time.

## Materials and methods

The URL http://merenlab.org/data/sar11-saavs contains a reproducible bioinformatics workflow that extends the descriptions and parameters of programs used here for (1) the metapangenome of SAR11 using cultivar genomes, (2) the profiling of metagenomic reads that the cultivar genomes recruited, (3) the analysis of single nucleotide variants using Deep Learning, and (4) the visualization of single nucleotide variants in the context of protein structures.

### SAR11 cultivar genomes

We acquired the genomic content of 21 SAR11 isolates from NCBI and simplified the deflines using anvi'o (*Eren et al., 2015*). We then concatenated all contigs into a single FASTA file, and generated an anvi'o contigs database, during which Prodigal (*Hyatt et al., 2010*) v2.6.3 identified open reading frames in contigs, and we annotated them with InterProScan (*Zdobnov and Apweiler, 2001*) v1.17. *Supplementary file 1b* reports the main genomic features.

### Metagenomic datasets

We acquired 103 metagenomes from the European Bioinformatics Institute (EBI) repository under the project IDs ERP001736 (n = 93; TARA Oceans project) and ERP009703 (n = 10; Ocean Sampling Day project), and removed noisy reads with the illumina-utils library (*Eren et al., 2013b*) v1.4.1 (available from https://github.com/meren/illumina-utils using the program 'iu-filter-quality-minoche' with default parameters, which implements the method previously described by *Minoche et al. (2011)*. *Supplementary file 1a* reports accession numbers and additional information (including the number of reads and environmental metadata) for each metagenome.

### Pangenomic analysis

We used the anvi'o pangenomic workflow (*Delmont and Eren, 2018*) to organize translated gene sequences from SAR11 genomes into gene clusters. Briefly, anvi'o uses BLAST (*Altschul et al., 1990*) to assess the similarity between each pair of amino acid sequences among all genomes, and

then resolves this graph into gene clusters using the Markov Cluster algorithm (*Enright et al., 2002*). We built the gene clustering metric using a minimum percent identity of 30%, an inflation value of 2, and a maxbit score of 0.5 for high sensitivity. Anvi'o used the occurrence of gene clusters across genomes data, which are also reported in *Supplementary file 1e*, to compute clustering dendrograms both for SAR11 genomes and gene clusters using Euclidian distance and Ward linkage algorithm.

## Estimating distances between isolate genomes based on full-length 16S ribosomal RNA gene sequences

We used the program 'anvi-get-sequences-for-hmm-hits' (with parameters '–hmm-source Ribosomal_RNAs' and '–gene-name Bacterial_16S_rRNA') to recover full-length 16S ribosomal RNA gene sequences from the anvi'o contigs database for the 21 isolate genomes. We then used PyANI (*Pritchard et al., 2016*) through the program 'anvi-compute-ani' to estimate pairwise distances between each sequence.

## Competitive recruitment and profiling of metagenomic reads

We mapped reads competitively from each metagenome against a single FASTA file containing all SAR11 genomes using Bowtie2 (*Langmead and Salzberg, 2012*) v.2.0.5 with default parameters, and converted the resulting SAM files into BAM files using samtools (*Li et al., 2009a*) v1.3.1. Competitive read recruitment ensures that short reads that match to more than one genome are assigned uniquely and randomly to one of the matching genomes. This minimizes computational biases at the mapping level and avoid inflated coverage statistics. To confirm our observations, we also used BWA (*Li and Durbin, 2009b*) to recruit reads (with the option n = 0.05). We used anvi'o to generate profile databases from the BAM files and combine these mapping profiles into a merged profile database, which stored coverage and variability statistics as outlined in *Eren et al. (2015)*. *Supplementary file 1c* reports the mapping results (number of recruited reads, as well as mean coverage and detection statistics) per genome across the 103 metagenomes.

## Determining the coverage of HIMB83 genes across metagenomes

The anvi'o merged profile database contains the coverage of individual genes across metagenomes. We normalized the coverage of HIMB83 genes in each metagenome (summarized in *Supplementary file 1j*) and calculated their coefficient of gene variation. We used the coefficient of gene variation estimates to identify metagenomes in which HIMB83 was well detected, yet the coverage values of its genes were highly unstable, which is an indicator of non-specific read recruitment from other lineages.

## Determining the main ecological niche and core genes of 1a.3.V

We considered metagenomes in which HIMB83 was sufficiently abundant (mean genomic coverage >50X) with a stable detection of its genes (coefficient of gene variation <1.25) to represent the main ecological niche of 1a.3.V. To determine the core 1a.3.V genes, we first disregarded metagenomes that displayed an unusually high coefficient of gene coverage variation (*Figure 1—figure supplement 1*), which can indicate non-specific read recruitments from other abundant populations. The 74 metagenomes fitting these criteria are summarized in *Supplementary file 1i*. We defined the subset of HIMB83 genes as the core 1a.3.V genes if in each of the 74 metagenomes, the mean coverage of a gene remained within a factor of 5 of the mean coverage across all genes. The 799 genes fitting this criterion are summarized in *Supplementary file 1j*.

## Calculation of percent identity distributions of recruited metagenomic short reads

We used percent identity distributions to broadly characterize how well short reads within a metagenome matched to the reference sequences by which they were recruited. We determined the percent identity for each read as $100 \times (N - n)/N$ where $n$ is the number of mismatches to the reference and $N$ is the read length. For simplicity, visualization of these distributions only included reads lengths of which matched to the median read length, and we defined bins to contain only one unique value. For example, if the median length of reads was 100, the bin domains for visualization

purposes were $(99, 100], (98, 99], (97, 98], \ldots, [0, 1]$. In contrast, all statistical calculations were carried out using all read lengths.

## Generating single-nucleotide variants (SNV) data

We used the program 'anvi-gen-variability-profile' to report variability tables describing the nucleotide frequency (i.e., ratio of the four nucleotides) in recruited metagenomic reads per SNV position. To study the extent of variation of the core 1a.3.V genes across all metagenomes, we instructed anvi'o to report positions with more than 1% variation at the nucleotide level (i.e., at least 1% of recruited reads differ from the consensus nucleotide). To compare the densities of SAAVs to SNVs, we instructed anvi'o to report only positions with more than 10% variation at the nucleotide level. *Supplementary file 1c* reports the density of SNVs for all SAR11 genomes across all metagenomes. We also used anvi'o to report SNVs for a subset of genes and metagenomes, and by considering only nucleotide positions with a minimum coverage cut-off across metagenomes under consideration. Controlling the minimum coverage of single nucleotide positions across metagenomes improves confidence in variability analyses. *Supplementary file 1L* reports the SNV density values for all core 1a.3.V genes.

## Definitions of 'SAAV', 'allele frequency' and 'AAST'

A single amino acid variant (SAAV) is a codon position that exhibits variation in a metagenome, and the unique identifier of a SAAV is a single codon position and a metagenome. The position of a SAAV in the reference sequence, and a vector of 21 elements that contain the allele frequencies of each amino acid as well as the stop codon fully characterize a SAAV. The allele frequency of an amino acid is equal to the number of short reads that fully cover the codon that resolves to the amino acid, divided by the total number of reads that fully cover the same position (the sum of all 21 allele frequencies is therefore 1). We also attributed to each SAAV an amino acid substitution type (AAST), which corresponds to the two amino acids with the largest and second largest allele frequencies.

## Generating single-amino acid variants (SAAVs) data

The program 'anvi-gen-variability-profile' (with an additional '–engine AA' flag) reported variability tables describing the allele frequencies for each SAAV. Anvi'o only considers short reads that cover the entire codon to determine amino acid frequencies at a given codon position in a metagenome. We instructed anvi'o to report only positions with more than 10% variation at the amino acid-level (i.e., at least 10% of recruited reads differ from the consensus amino acid). *Supplementary file 1L* reports the density of SAAVs for all SAR11 genome across all metagenomes. We also used anvi'o to report SAAVs for a subset of genes and metagenomes, and by considering only gene codons with a minimum coverage cut-off of 20X across all metagenomes of interest. Controlling the minimum coverage of gene codons across metagenomes improves confidence in variability analyses.

## Differential occurrence of amino acids in SAAVs and in the core 1a.3.V genes

We determined the amino acid composition in the 799 core 1a.3.V genes as well as in SAAVs maintained in each metagenome using anvi'o programs 'anvi-get-aa-counts' and 'anvi-get-codon-frequencies' (with the flag '–return-AA-frequencies-instead'). We quantified the amino acid composition of all core 1a.3.V genes of in HIMB83 using the program 'anvi-get-aa-counts'. In contrast, we quantified the amino acid composition of SAAVs by calculating the frequency of a given amino acid being one of the two dominant alleles. We then calculated p-values via a binomial test that represents the probability of observing the difference between amino acid frequencies computed over all core 1a.3.V genes versus only 1a.3.V SAAVs, given the null hypothesis that amino acids in 1a.3.V SAAVs are distributed according to the same distribution as the amino acids in the core 1a.3.V genes.

## Estimating a neutral AAST frequency distribution

This calculation provides an estimate for the AAST frequency distribution given strictly neutral mutations. Unlike the neutral theory of evolution, it excludes the influential effects of purifying selection

(negative selection coefficients). Since all mutations are equally likely to drift to detectable frequencies under a neutral model, the expected number of variant positions that have $C_i$ and $C_j$ as their two dominant alleles, is proportional to the rate that $C_i$ mutates to $C_j$ plus the rate that $C_j$ mutates to $C_i$. Expressed mathematically,

$$E\left(N_{\{C_i, C_j\}}\right) \propto P(C_i|m)P(C_i \rightarrow C_j|C_i,m) + P(C_j|m)P(C_j \rightarrow C_i|C_j,m)$$

Where $E\left(N_{\{C_i, C_j\}}\right)$ is the expected number of variant positions that have $C_i$ and $C_j$ as their two dominant alleles, $P(C_i|m)$ is the probability that a $C_i$ position mutates given that a mutation has occurred, and $P(C_i \rightarrow C_j|C_i,m)$ is the probability that such a mutation will mutate to $C_j$. Assuming all sites are equally likely to mutate, $P(C_i|m)$ is equivalent to the fraction of codons in the reference sequence that are $C_i$, and we denote this quantity as $f_{C_i}$. To extend the equation to the expected number of variant positions that have amino acids $A_1$ and $A_2$ as their two dominant alleles, that is a quantity proportional to the AAST frequency, one must enumerate over all codons in $A_1$ and $A_2$:

$$\mathbb{E}\left(N_{AAST=\{A_1, A_2\}}\right) \propto \sum_{C_i \in A_1} \sum_{C_j \in A_2} P(\{C_i, C_j\})$$

In general, $P(C_i \rightarrow C_j|C_i,m)$ will depend primarily upon the nucleotide edit distance between $C_i$ and $C_j$, which we denote as $d$, as well as the transition/transversion rate ratio, which we will denote $\kappa$. How the model handles these aspects will critically influence the expected frequency distribution. To encapsulate the broadest possible interpretation of the neutral model, we evaluate expressions for two extreme cases: In the first case (Model 1), we assume that the probability of an edit distance $d > 1$ is 0 (in reality, estimates at least for eukaryotes range from 0.003 [*Smith et al., 2003*] to 0.03 [*Schrider et al., 2011*]). We also impose a $\kappa$ value of 2 so that transitions are twice as likely as transversions. Intuitively, these impositions have the effect of skewing the AAST frequency distribution towards AASTs that possess highly similar codons. In the second case (Model 2), we assume all codon transitions are equally likely regardless of edit distance or the number of transitions/transversions ($\kappa = 1$). Intuitively, this has the effect of homogenizing the AAST frequency distribution towards a more uniform-like distribution.

In Model 1, $P(C_i \rightarrow C_j|C_i,m) = \frac{1}{3}\delta_{d,1}P(m)$, where $\delta_{d,1}$ is a Kronecker delta function describing the probability the mutation has an edit distance $d$, 1/3 is the probability that the correct nucleotide position is mutated, and $P(m)$ is the probability that the mutation occurs based on whether or not it is a transition. Formally,

$$P(m) = \begin{cases} \kappa/\kappa + 2; & m = transition \\ 1/\kappa + 2; & m = transversion \end{cases}$$

In Model 2, $P(C_i \rightarrow C_j|C_i,m) = 1/63$, since all 63 possible mutations are permissible and equally probable. The expressions for $E\left(N_{AAST=\{A_1, A_2\}}\right)$ for Model 1 and Model 2 thus simplify to:

$$^{M1}\mathbb{E}\left(N_{AAST=\{A_1, A_2\}}\right) \propto \sum_{C_i \in A_1} \sum_{C_j \in A_2} \langle f_{C_i}, f_{C_j} \rangle \delta_{d,1} P(m)$$

$$^{M2}\mathbb{E}\left(N_{AAST=\{A_1, A_2\}}\right) \propto \sum_{C_i \in A_1} \sum_{C_j \in A_2} \langle f_{C_i}, f_{C_j} \rangle$$

where $M1$ and $M2$ refer to Model 1 and Model 2, respectively. To compare directly with observation, we extracted $f_{C_i}$ for the 64 codons from the HIMB83 reference sequence using 'anvi-get-codon-frequencies' and the distributions under both models were calculated from the above equations.

## Predicting 3D structure of proteins using template-based modeling

We used a template-based structure modeling tool, RaptorX Structure Prediction (*Källberg et al., 2012*), to predict structures of 1a.3.V amino acid sequences based on available data from the Protein Data Bank (PDB) (*Bernstein et al., 1977*). We used the program blastp in NCBI's BLAST distribution to identify core 1a.3.V genes that matched to an entry with at least 30% similarity over the

length of the given core gene. We then programmatically mapped SAAVs from metagenomes onto the predicted tertiary structures, and used PyMOL (*DeLano, 2002*; *Schrödinger LLC, 2015*) to visualize these data. We colored SAAVs based on RaptorX-predicted structural properties, including solvent accessibility and secondary structure.

## Identifying genes with disproportionately high number of temperature-correlated positions

First, we calculated the number of temperature-correlated and temperature-uncorrelated positions for each of the 1a.3.V core genes. Then, we performed a one-sided binomial test that these numbers are biased towards higher proportion of temperature-correlated positions compared to a model distribution defined from the total number of temperature-correlated positions in 1a.3.V. Since there were 4,592 such positions out of 37,416, the model probability of success was defined as $p_0 = \frac{4592}{37416} = 0.123$. In other words, the expected proportion of variant positions in a gene that are temperature-correlated is 0.123 under the model. We corrected the resulting p-values for each gene for multiple testing using Benjamini & Hochberg's method (*Benjamini and Hochberg, 1995*).

## Predicting transmembrane, periplasmic, and cytosolic regions in the glycine betaine permease

To categorize amino acid positions as transmembrane, periplasmic, and cytosolic, we used Phobius (*Käll et al., 2004*; *Käll et al., 2007*), a membrane topology and prediction software through the webserver at http://phobius.sbc.su.se. The output is a probability of the four classes for each residue, and to simplify the data we categorized each residue into the class found to be most probable. We removed residues with signaling peptide association from downstream analyses.

## Recovering codon position-specific AASTs from SAAVs

We simplified the hyper-dimensional SAAV data into a simpler presence-absence matrix for downstream analyses. For this, we defined codon position-specific AASTs (cAASTs) and summarized their occurrence across metagenomes. In such a table the value of '1' indicates that a given metagenome had a SAAV at a given codon position that resolved to a given AAST. In contrast, the value '0' indicates that the metagenome did not have a SAAV that resolved to this AAST. In the latter case a given metagenome may have another AAST in this particular codon position (in which case this information would appear in another row in the same table that is affiliated with the same AAST with the same codon position). Hence, each AAST listed in the first column of the table will be unique to a single codon position, yet a given codon position may have different AASTs in different metagenomes, resulting in multiple AASTs in the resulting table that belong to the same codon position. Combining AAST with the codon position would then result in a unique cAAST.

## Application of deep learning to codon-position-specific AASTs data

To estimate an unbiased distance between our metagenomes based on SAAVs, we used a novel deep neural network modification called the auto-clustering output layer (ACOL). Briefly, ACOL relies on a recently introduced graph-based activity regularization (GAR) technique for competitive learning from hyper-dimensional data to demarcate fine clusters within user-defined 'parent' classes (*Kilinc and Uysal, 2017*). In this application of ACOL, however, we modified the algorithm so it can reveal latent groups in our SAAVs in a fully unsupervised manner through frequent random sampling of SAAVs to create pseudo-parent class labels instead of user-defined classes (*Kilinc and Uysal, 2018*). See the URL http://merenlab.org/data/sar11-saavs for the details of the pseudo parent-class generation algorithm, and the reproducible distance estimation workflow in Python.

## Other statistical tests and visualization

We used the aov function in R to perform one way ANOVA tests, used the ggplot2 (*Ginestet, 2011*) package for R to visualize the relative distribution of 1a.3.V genes and geographic distribution of proteotypes, and finalized all figures using an open-source vector graphics editor, Inkscape (available from http://inkscape.org/).

## Code and data availability

The vast majority of analyses relied on the open-source software platform anvi'o v2.4.0 (available from http://merenlab.org/software/anvio). The URL http://merenlab.org/data/sar11-saavs serves the remaining custom code used in our analyses. We made available (1) SAR11 isolate genomes (doi:10.6084/m9.figshare.5248945), (2) the anvi'o contigs database and merged profile for SAR11 genomes across metagenomes (doi:10.5281/zenodo.835218) and the static HTML summary for the mapping results (doi:10.6084/m9.figshare.5248453), (3) the SAR11 metapangenome (doi:10.6084/m9.figshare.5248459), single-nucleotide and single-amino acid variant reports for 1a.3.V across 74 TARA Oceans metagenomes (doi:10.6084/m9.figshare.5248447), and (4) SAAVs overlaid on predicted tertiary structures of 58 core 1a.3.V genes (doi:10.6084/m9.figshare.5248432). The URL http://anvi-server.org/p/4Q2TNo serves an interactive version of the SAR11 metapangenome, and the URL http://data.merenlab.org/sar11-saavs serves an interactive web page to investigate the link between SAAVs and predicted protein structures.

## Acknowledgements

We thank the TARA Oceans consortium for generating metagenomic datasets of great legacy, as well as all researchers involved in the characterization of SAR11 isolates. We thank Kostas Konstantinidis, Edward Delong, Lois Maignien, Mike Lee, and the members of the Meren Lab for helpful discussions. We acknowledge the support of the Natural Sciences and Engineering Research Council of Canada (NSERC) to EK, as well as the Frank R Lillie Research Innovation Award and the start-up funds from the University of Chicago to AME.

## Additional information

### Funding

| Funder | Author |
| --- | --- |
| University of Chicago | A Murat Eren |
| Marine Biological Laboratory | A Murat Eren |

The funders had no role in study design, data collection and interpretation, or the decision to submit the work for publication.

### Author contributions

Tom O Delmont, Conceptualization, Data curation, Formal analysis, Validation, Investigation, Visualization, Methodology, Writing—original draft; Evan Kiefl, Conceptualization, Data curation, Software, Formal analysis, Validation, Investigation, Visualization, Methodology, Writing—review and editing; Ozsel Kilinc, Software, Writing—review and editing; Ozcan C Esen, Software, Visualization, Methodology; Ismail Uysal, Software, Supervision; Michael S Rappé, Supervision, Validation, Writing—review and editing; Steven Giovannoni, Supervision, Validation, Project administration, Writing—review and editing; A Murat Eren, Conceptualization, Resources, Data curation, Software, Formal analysis, Supervision, Funding acquisition, Validation, Investigation, Visualization, Methodology, Writing—original draft, Project administration, Writing—review and editing

### Author ORCIDs

Tom O Delmont (ID) http://orcid.org/0000-0001-7053-7848
Evan Kiefl (ID) https://orcid.org/0000-0002-6473-0921
A Murat Eren (ID) https://orcid.org/0000-0001-9013-4827

### Decision letter and Author response

Decision letter https://doi.org/10.7554/eLife.46497.040
Author response https://doi.org/10.7554/eLife.46497.041

# Additional files

## Supplementary files

• Supplementary file 1. Details of SAR11 genomes, marine metagenomes, and metagenomic read recruitment results. (**a**) Summary of 103 Tara Oceans and Ocean Sampling Day metagenomes. (**b**) Features of 21 SAR11 isolate genomes, clades to which they belong, and their genome-level metagenomic read recruitment summaries. (**c**) Comprehensive summary of metagenomic read recruitment results per SAR11 genome, including the number of recruited reads, mean coverage, relative distribution, as well as the detection of 21 SAR11 genomes across 103 metagenomes, along with the total number of single nucleotide variants (SNVs) and single amino acid variants (SAAVs) identified in each metagenome. (**d**) Summary of the SAR11 metapangenome and gene cluster membership statistics. (**e**) Presence/absence summary of gene clusters across SAR11 genomes. Distances between SAR11 genomes based on (**f**) their full-length 16S ribosomal RNA genes and (**g**) whole genome average nucleotide identity. (**h**) Relative distribution and abundance of 21 SAR11 genomes, 31 *Prochlorococcus* genomes, and 957 additional marine population genomes across from three studies across 103 metagenomes. (**i**) SAR11 genomes with more than 50X coverage across Tara Oceans Project and Ocean Sampling Day metagenomes. (**j**) Detection of individual HIMB83 genes across metagenomes and whether they belong to 1a.3.V core or not. (**k**) Summary of the degree of correlation between percent identity histograms of metagenomics reads recruited by HIMB83 and environmental data reported per metagenome. (**l**) SNV and SAAV density of HIMB83 across metagenomes. This table also reports the predicted functions of individual HIMB83 genes and their DNA sequences, and functional summary of HIMB83 genes that represent core 1a.3.V genes and those that belong to environmental accessory genes of HIMB83.
DOI: https://doi.org/10.7554/eLife.46497.018

• Supplementary file 2. Details and raw data for single-nucleotide variants, single-amino acid variants, and amino acid substitution types. (**a**) SNV density of core 1a.3.V genes, (departure from consensus of >1%). (**b**) SNV density of core 1a.3.V genes (departure from consensus of >10%) (**c**) SAAV density of core 1a.3.V genes (departure from consensus of >10%). (**d**) Number of SAAVs among core 1a.3.V genes. (**e**) Distribution of invariant core 1a.3.V proteins across metagenomes. (**f**) Frequency and proportion of amino acids in SAAVs. (**g**) Comprehensive summary statistics and ratios for amino acid substitution types (AASTs) across metagenomes. (**h**) Unique coordinates per AAST in 738,324 core 1a.3.V genes (cAASTs) (1) that were covered more than 20X across 74 metagenomes, (2) and in which a divergence >10% from consensus was observed in the frequency of amino acids. (**i**) cAASTs across metagenomes. This table also reports BLOSUM estimates per AAST, and BLOSUM AAST summary statistics across core 1a.3.V genes.
DOI: https://doi.org/10.7554/eLife.46497.019

• Supplementary file 3. Relationships between SAAVs and protein structures and temperature. (**a**) SAAV characteristics, including solvent accessibility for each SAAV belonging to a core 1a.3.V gene with a successfully predicted protein structure. (**b**) The correlation of allele frequencies to temperature for each position containing at least one SAAV. (**c**) The summary of the proportion of temperature-correlated and temperature-uncorrelated SAAV positions per core 1a.3.V gene.
DOI: https://doi.org/10.7554/eLife.46497.020

• Supplementary file 4. Details of proteotypes within 1a.3.V. (**a**) Deep Learning-estimated distances between 74 metagenomes based on cAASTs reconstructed from 738,324 SAAVs. (**b**) Six 1a.3.V proteotypes. (**c**) Summary of all data per metagenome to estimate significant determinants of proteotype organization. (**d**) ANOVA test statistic per sample data category given the six proteotypes.
DOI: https://doi.org/10.7554/eLife.46497.021

• Transparent reporting form
DOI: https://doi.org/10.7554/eLife.46497.022

## Data availability

The vast majority of analyses relied on the open-source software platform anvi'o v2.4.0 (available from http://merenlab.org/software/anvio). The URL http://merenlab.org/data/sar11-saavs serves the remaining custom code used in our analyses. We made available (1) SAR11 isolate genomes (doi:10.6084/m9.figshare.5248945), (2) the anvi'o contigs database and merged profile for SAR11 genomes

across metagenomes (doi:10.5281/zenodo.835218) and the static HTML summary for the mapping results (doi:10.6084/m9.figshare.5248453), (3) the SAR11 metapangenome (doi:10.6084/m9.figshare.5248459), single-nucleotide and single-amino acid variant reports for 1a.3.V across 74 TARA Oceans metagenomes (doi:10.6084/m9.figshare.5248447), and (4) SAAVs overlaid on predicted tertiary structures of 58 core 1a.3.V genes (doi:10.6084/m9.figshare.5248432). The URL http://anvi-server.org/p/4Q2TNo serves an interactive version of the SAR11 metapangenome, and the URL http://data.merenlab.org/sar11-saavs serves an interactive web page to investigate the link between SAAVs and predicted protein structures.

The following datasets were generated:

| Author(s) | Year | Dataset title | Dataset URL | Database and Identifier |
|---|---|---|---|---|
| A Murat Eren | 2017 | Anvi'o split profile for HIMB83 across metagenomes | https://doi.org/10.6084/m9.figshare.5248435 | figshare, 10.6084/m9.figshare.5248435 |
| A Murat Eren | 2017 | The SAR11 Metapangenome | https://doi.org/10.6084/m9.figshare.5248459 | figshare, 10.6084/m9.figshare.5248459 |
| A Murat Eren | 2017 | Anvi'o summary of SAR11 genomes across metagenomes | https://doi.org/10.6084/m9.figshare.5248453 | figshare, 10.6084/m9.figshare.5248453 |
| A Murat Eren | 2017 | Raw SNV and SAAV data for SAR11 1a.3.V | https://doi.org/10.6084/m9.figshare.5248447 | figshare, 10.6084/m9.figshare.5248447 |
| A Murat Eren | 2017 | S-LLPA SAAVs | https://doi.org/10.6084/m9.figshare.5248432 | figshare, 10.6084/m9.figshare.5248432 |
| A Murat Eren | 2017 | Anvi'o merged profile database for 21 SAR11 isolates across metagenomes | https://doi.org/10.5281/zenodo.835218 | Zenodo, 10.5281/zenodo.835218 |

The following previously published datasets were used:

| Author(s) | Year | Dataset title | Dataset URL | Database and Identifier |
|---|---|---|---|---|
| Shinichi Sunagawa et al. | 2015 | Ocean plankton. Structure and function of the global ocean microbiome. | https://www.ncbi.nlm.nih.gov/sra/?term=PRJEB1787 | NCBI SRA, PRJEB1787 |
| Anna Kopf et al. | 2015 | The ocean sampling day consortium | https://www.ebi.ac.uk/ena/data/view/PRJEB5129 | EMBL-EBI, PRJEB5129 |

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
