## [Decision Letter]

Thank you for submitting your work entitled "The global biogeography of a single SAR11 population is governed by natural selection" for consideration by *eLife*. Your article has been reviewed by three peer reviewers, including Paul Rainey as the Reviewing Editor, and the evaluation has been overseen by Detlef Weigel as the Senior Editor. The outside reviewers have opted to remain anonymous.

Our decision has been reached after extensive consultation between the reviewers. Based on these discussions and the individual reviews below, we regret to inform you that your work will not be considered further for publication at this time. However, we do see the potential for worthy *eLife* publication, provided all the issues raised by the reviewers can be addressed. Any resubmission would need to be a very substantial revision and would be treated as a new submission.

All reviewers agree that the topic is of considerable interest. They also indicate that your methods appear to have many strengths, but at the same time all are in agreement that much more needs to be done to justify your approaches. For example, Reviewer 3 requests calibration against standard population genetic approaches. Additional problems concern the main claims of the paper such as selective sweeps, the effect of selection, and so forth, which are stated, but not evidently backed by data. Further issues surround use and understanding of terms in population biology/ecology: Reviewer 2 points out the problems with refereeing to diversification within a population when the "population" of interest is really a substantial portion of a genus.

*Reviewer #1:*

This deals with the causes of patterns of diversity in a widespread marine microbe. The data come from a set of contigs assembled from shot gun sequencing of 21 genomes onto which are read-mapped metagenomic data from 103 samples. The focus is a particular clone that is particularly abundant in southern samples. A core genome is assembled from the metagenomic data and this is interrogated for signatures of selection using a curious approach based on physico-chemical properties of amino acids. A further step looks for correlations with water currents.

At first glance I found the paper reasonably clear, but the closer I read, the more murky things became. In fact I am not too sure what the authors have really done and I am not persuaded that their main claim, that selection is responsible for biogeography in S-LLPA, is supported by their data. Overall the level of explanation is insufficient to allow the reader to understand what has been done.

Among the difficulties:

Title/Abstract: The title states that biogeography is governed by selection, but in the Abstract the authors use more temperate language. So I guess they are not so sure. The Abstract claims that systematic purifying selection and adaptive mechanisms governing non-synonymous variation have been identified, but at best what they have are signatures consistent with expectations. I disagree that analysis of sequence data reveals different niches (it may reveal the existence of ecotypes that may be indicative of distinct niches). What is a proteotype?

Place of isolation of the focal 21 SAR11 genomes is important in a study of biogeography, but no attention has been given to this.

There are many challenges in drawing conclusions from metagenomic data via read-mapping to a reference. Particularly problematic is the fact that diversity of closely related organisms within a sample is not known. Thus when many SNPs are detected is this indicative of a diverse population, or is it a signature that comes from a single over-represented clone that is rather divergent? Similarly, the authors take high similarity as signature of a recent sweep, but this might reflect a single over-represented genome from a closely related organism. I'm not sure how the authors dealt with these and similar problems.

With regard to selective sweeps, the results of the analysis appear not to be given (subsection “SNVs to SAAVs: Accurate characterization of non-synonymous variation”, gives the conditions that need to be fulfilled in order for a protein to have undergone a sweep, but not the results).

Patterns of amino acid substitution types seems an interesting way to go, but this needs to be unpacked, explained and justified.

Partitioning of SAAV between warm and cold currents: there seems to be a correlation with clustering based on SAAVs, but as above, I need to have more explanation. Much relies on the robustness of the SAAV profile clustering (Figure 3A) and there is no indication of its statistical likelihood.

The crunch of the paper is rejection of the neutral model, but this is done by simply saying that neutral models could not explain the observed patters of amino acid substitutions, but no evidence is provided. Also, Figure 4, which I hoped would enlighten me, does not exist.

In general I found the figures overly dense and very difficult to understand given the paucity of explanation in the captions. Take Figure 1: latitudinal gradient, I presume north is at the top? Colouring seems inconsistent and so on.

*Reviewer #2:*

It's hard to tell who this paper is written for. The bioinformatics applications are the most interesting: identifying the core genome of a lineage of interest from metagenomics data, using a novel algorithm for (somehow) identifying significant subclades within the lineage, and finding a surprisingly high frequency of certain amino acid substitutions that are not predicted by Blosum. There are also interesting implications for ecological diversification, but this aspect of the paper is written without enough explanation or interpretation of data, as I will detail. Also, I think most microbial ecologists would be amazed to see a paper on diversification within "a single population" that is actually not a population at all, but rather a fair chunk of a genus.

It is so inappropriate to call the S-LLPA lineage a "single population," since it has nearly 20% sequence divergence within it, and because there is clearly profound ecological diversification within this group. (Given that the Eren group has made tremendous progress toward discovering newly divergent, ecologically distinct populations of bacteria with their MED algorithm, it seems ironic that the group is now calling this phylogenetically huge group a single population.) Calling this group a population seems to have led to the conclusion that "their broad geographic prevalence suggest<s> dispersion is not a limiting factor for SAR11 in surface seawater…." But this conclusion is based on a false reification of all the various true populations within what the authors are calling a population.

The authors talk about S-LLPA offers "a unique opportunity to study the genomic diversity and evolutionary genomics of a single marine microbial population…." Not so unique when the "population" is a huge chunk of a genus.

The authors do justify their calling this clade a population because it constitutes what is recruited by a single isolate from various metagenomes. But their own work clearly shows, to my mind, that this is not a good way to discover populations.

The authors have used their Deep Learning algorithm for demarcating six phylogenetic groups, which they call proteotypes, from their S-LLPA lineage. It is frustrating that the authors neglected to give any rationale for their approach. Is it based on sequence clustering of shared genes? Is it based on genome content sharing? Or both? (Even though the algorithm has been published, they should say something briefly about what the inputs are and how it works.)

On two occasions the authors write about protein sweeps. They state that the sweeps are "rare." I think most evolutionary ecologists of bacteria would want to know how rare, and in which genes. Also, it would be important to indicate whether the sweeps traveled across ecologically distinct groups. In the first paragraph of the Discussion, the authors mention that there are more sweeps in warm currents, but this should be fleshed out and made clearer.

Subsection “Purifying selection governs the identity of amino acid substitution types”, first paragraph. It wasn't clear to me whether the authors were claiming that their focus group was unique in having a stable proportion of amino acids on a global scale, or whether other groups are showing the same pattern.

In the subsection “Purifying selection governs the identity of amino acid substitution types”, I don't see how the authors are finding evidence of purifying selection. Rather, it seems they are finding the constraints on adaptive substitutions of amino acids. That's different from purifying selection, which is usually the evolutionary stability of a particular sequence.

It's interesting that there are two clades within S-LLPA that are associated with different temperature regimes. And it's interesting that the subclades within these clades are different in the geographic regions where they are most abundantwith similar temperatures. The interpretation of this result depends on whether different cold-adapted subclades are never found together (meaning that they may have no adaptation differences) or whether they sympatric to some extent (meaning that they are coexisting based on ecological differences other than temperature partitioning). It seems that the latter is the case, but the authors should have made this clearer. For example, pie charts showing the relative abundances of the subclades in each region would be useful.

*Reviewer #3:*

The work by Delmont et al. describes a large-scale and thorough analysis of an important marine bacterium population, SAR111. They use multiple approaches to identify signals of selection to understand the evolution and ecology of these microbes. The authors deserve credit for their very clear presentation of data and methods. However, I have multiple concerns regarding the analysis and interpretation of the data:

1) The nature, and strength of, selection pressures acting on this bacterial population are not clearly defined.

a) The description of the substitution analysis "suggests a powerful influence of purifying selection", while also "suggesting a role for adaptive processes for functional diversification of S-LLPA proteins". Which of these dominates? Could this just be noisy neutral evolution – what is the expected variation of the S statistic, and the SAAV patterns, under a neutral scenario?

b) The neutral model of Hellweger et al. still predicts geographic separation of strains, which should be made clear in the Introduction. A falsification of this would be a repeated association of bacterial strains with ecological niche. This appears to be the case at the highest level of the hierarachy inferred with "deep learning", but at lower levels, predictors are genetic, not ecological, and therefore likely represent characteristics that were identified by the clustering algorithm itself. This might be expected under a neutral model.

2) The authors have opted against traditional population genetics methods. It is possible that the new methods are a substantial improvement, but they need to be justified. Some examples:

a) The authors justify using the "S" statistic rather than d(N)/d(S) based on there being many instances of multiple substitutions within the same codon. But this is exactly what d(N)/d(S) is meant to correct for, and S does not capture – there may be many synonymous substitutions on an evolutionary pathway, but if only one of them is non-synonymous, then S will only count this only as a non-silent change, underestimating purifying selection. What is the neutral expectation for S?

b) The BLOSUM matrices are used as "as a proxy to assess the functional difference between the pairs of amino acids", which are then compared to the substitution frequencies – but this is circular reasoning, as these matrices are calculated from substitution frequencies. The versions used in the paper are also inappropriate for this dataset, as they were calculated using more closely-related sequences than are compared in this paper.

c) "Deep learning" is applied to cluster populations together through a method designed without genetic data in mind. This does not mean it is wrong, but as it is not an intuitive approach, it would be helpful to see validation against a more standard method, such as the fixation index, F(ST). This would help understand the distinct properties of the six subclusters – for instances, proteotype D looks to be a single successful "strain" – is this expected to be picked up as a signal by the "deep learning"?

3) Quality of read mapping. Many of the sequences being compared are highly divergent, but most mapping software is intended to align reads to highly similar reference sequences. While there is a promising correlation between the SNP densities identified by BWA and BOWTIE2, there is a greater than two-fold difference in the absolute densities. This suggests a very high false positive or false negative rate, depending on the method. The authors need to validate the performance of these algorithms when using highly divergent reads. Additionally, what is the justification for using the "the base-5 logarithm of the mean coverage of a gene remained within {plus minus}1 of that for the mean coverage across all genes" for assessing consistency of coverage?

[Editors’ note: what now follows is the decision letter after the authors submitted for further consideration.]

Thank you for submitting your article "Single-amino acid variants reveal evolutionary processes that shape the biogeography of a global SAR11 lineage" for consideration by *eLife*. Your article has been further reviewed by three peer reviewers, including Paul B Rainey as the Reviewing Editor and Reviewer #1, and the evaluation has been overseen by Detlef Weigel as the Senior Editor. The following individual involved in review of your submission has agreed to reveal his identity: Fred Cohan (Reviewer #2).

The reviewers have discussed the reviews with one another and the Reviewing Editor has drafted this decision to help you prepare a further revised submission.

All reviewers appreciate and acknowledge the significant work that has gone into the paper during the course of the revision, the reviewers are all equally enthusiastic about the paper. However, there remain issues that need attention in order for the paper achieve the impact that it deserves. The reviewers' comments are pasted below, with the collective comments boiling down to a request to supplement the deep learning analysis with a more standard evolutionary approach in order for the reader to know whether the results are consistent with what most people in the field know. Here it is important to clearly distinguish between neutral evolution, purifying selection, and adaptive divergence. It is the latter that is particularly relevant to addressing the question of the possible ecological distinctions among populations.

Collected reviewer comments:

The paper is vastly improved since the original version. It is now much clearer and more compelling, and the paper deals with the issue of populations better.

The paper is fascinating for showing that whatever cohesion occurs at >95% ANI in pretty much every other bacterial group does not occur here in SAR11. The paper is also really interesting for giving a robust and sensitive way to identify ecologically distinct metagenome groups from sequence data.

My concerns are mostly focused on what the authors have found out about ecological diversity within the 1a.3.V lineage.

First, I appreciate that the authors are trying to be more careful about using the concept of population, but there are still places where they are still unclear about what they mean. The authors write of "remarkable intra-population genomic diversity…." It's not clear exactly what the authors mean by "population" here, but in this context what they mean is very important. If there isn't a good definition of population being used, then how can one talk about one population's intra-population genomic diversity being higher than another? It all depends on what you mean by population, and that's not clear here.

Figure 2 purports to show that there are not sequence-discrete populations within 1a.3.v, at least not in the sense that Kostas Konstantinidis demonstrates them in his studies, most recently in Jain et al., 2018. It is really interesting that all the SAR11 clades (with just this one clade shown in Figure 2) do not show the drop-off in recruitment of metagenome sequences below about 95% ANI. The authors argue that part of the great sequence diversity of what is recruited into this lineage (that is with much lower ANI) emerges from there being ecological diversity within the lineage (adaptations to different temperatures among different sublineages). I don't think this is a reasonable explanation, as almost every species taxon, nearly all obeying the 95% ANI drop-off rule, is also ecologically heterogeneous. I have suggested, in an interpretation of the recent Jain et al. paper, that the cohesion holding a species taxon together stems from an ability of recombination to act frequently at >95% ANI, plus the possibility of adaptive genes passing between ecologically distinct populations that are >95% ANI (Cohan, 2019) I'd argue that there is more potential for either recurrent recombination to limit divergence among ecologically distinct populations in SAR11, or greater opportunities for a single gene to be adaptive across different populations.

I'd like to add that whatever cohesion occurs at >95% ANI in most species taxa does not address the ecological diversity within species taxa, or whether there are sequence-discrete populations within each taxon. In fact, most species taxa have sequence-discrete and sequence-discoverable ecologically distinct populations. (This is the point of the second half of my Current Biology dispatch I mentioned.) So, I don't think that Figure 2 makes an argument that there are not sequence-discrete, ecologically distinct populations within the focus lineage.

The authors have noted that there is a small minority of genes that are invariant at the amino acid level across a given metagenome. In the earlier version, they interpreted these observations as evidence of sweeps, but here they have chosen not to address the dynamics that can be inferred by these invariant genes. I think that such single-gene sweeps are very interesting, and have been the topic of discussion of some very interesting papers, including papers by Jesse Shapiro and by Rex Malmstrom. I'd encourage the authors to bring this discussion back. I'll mention that these instances provide evidence that a generally adaptive gene has passed by recombination across all the ecologically distinct populations within a lineage, but I wouldn't expect the authors to necessarily buy in to that interpretation.

I found the authors' partitioning of metagenomes into ecologically distinct groups fascinating. And it's particularly interesting that these groups could not be revealed (at least entirely) by analyzing single reads (as shown in Figure 4—figure supplement 3). I'll encourage the authors to venture from their interpretation regarding whole metagenomes to inferring that there appear to be multiple ecologically distinct populations that cause these metagenomes to hold different niches. What might be the ecological distinctions of the populations? Clearly they are different for their temperature adaptation, as revealed by Figure 4 (mislabeled Figure 5). Perhaps the authors could glean something about what is different about the constituent populations from their Figure 4—figure supplement 3, that is by comparing the clusters that they do find with known environmental parameters.

Re the concluding statement about "everything is everywhere," I'd temper that by saying 'at least for marine bacteria'. This is because the statement does talk about "everything," so I think there needs to be some limitation to the conclusion.

Concerns about evolutionary analyses:

In response to point (1), the authors state:

"Simply, we observed that certain amino acids (mostly hydrophilic), and most notably, few AASTs (e.g., alanine/isoleucine) predominated in our SAAV table."

It has been well-established for decades that hydrophobic amino acids in solvent-inaccessible positions are more conserved (e.g. see reviews cited in the Introduction to Ramsey et al. 2011 Genetics 188(2):478-88). Reproducing this observation in the SAR11 dataset simply confirms that purifying selection can be observed over long timescales. It does not help identify what the changes are that enable ecological adaptation.

Delmont et al: "These points are interesting, but they do not favor natural selection over neutral evolution. Instead they provide information regarding permissible versus non-permissible diversification"

I assume "permissible" refers to purifying selection? Which is an example of natural selection – rather than random, neutral diversification. It is distinct from the adaptive/Darwinian evolution that allows the ecological differentiation of the bacteria. Not separating neutrality, purifying selection and adaptive evolution plagues the whole manuscript. The authors have a strain distribution that appears to reflect ecological differentiation, rather than neutral diffusion. They have a mutation distribution that mainly seems to reflect purifying selection, rather than neutral evolution. Some analyses (e.g. below) suggest adaptive evolution, which is potentially very interesting, but the overall trend is consistent with purifying selection. The authors need to clearly distinguish their different conclusions.

Delmont et al: "we determined that thousands of 1a.3.V allele frequency trajectories correlated with in situ temperature, in line with the biogeography of proteotypes"

It is not surprising that many of the variable sites correlate with temperature, because the proteotypes are defined using the variable sites, and the proteotypes correlate with temperature. The authors should seek to synthesize these data – do the variable sites have properties that suggest selection between niches? Do they concentrate in particular genes that might drive adaptation to different niches? Or do they simply correlate with the population structure?

Overall, I return to my original point from the first round of reviews: is the diversification mainly neutral; dominated by purifying selection; or dominated by adaptive evolution? The most interesting signals may not be from the dominant evolutionary process, of course.

In response to point 2, the authors state:

Delmont et al: "As far as we know, there is no published study that effectively takes into account multiple SNVs per codon without calculating the exact codon frequencies to estimate synonymity accurately…SNV density was high for 1a.3.V and many SNVs co-occurred in the same codon, rendering classical d(N)/d(S) analyses limited, which had never been challenged with extremely complex environments, if not completely irrelevant."

Yang (2007) MBE 24(8):1586-91 – with over 6,000 citations – describes how baseml can be used to calculate d(N)/d(S) using base substitution, rather than codon substitution, models. This accounts for multiple substitutions per codon, and provides a statistical test for evidence of selection versus neutrality, unlike the methods presented in the current version. It does not matter how complex the environments are – indeed, d(N)/d(S) has been applied to the question of SAR11 adaptation to different ecologies by Brown et al., 2012 and Luo and Hughes, 2012 Mol. Syst. Biol. 8:625. The authors should at least apply standard methods to the assembled genomes, based on the ecological separation they have determined. Methods also exist for calculation of this statistic from metagenomic data, and might be more informative.

Delmont et al: "The significant shift in SAAVs/SNVs ratio between cold currents and warm currents is of relevance to our study, especially in light of other insights (especially, the linkage between allele frequency trajectories and in situ temperature, and the biogeography of proteotypes), favors the predominant role of natural selection acting on the permissible amino acid diversification traits of this global SAR11 lineage."

The correlation between allele frequencies and temperature, and proteotype and temperature, is confounded by the correlation between proteotypes and allele frequencies. This should be explained (if this is wrong, please also make that clear in the text). The authors need to make clear whether the natural selection they are referring to is stronger/weaker purifying selection changing the SAAV/SNV ratio; or positive selection driving adaptive diversification increasing the SAAV/SNV in some populations.

Delmont et al: "Regarding the fixation index, we agree with the reviewer that this would have been a valuable addition to our work…We believe the significant amount of methodological development in this work that offers descriptions of the biogeography of the most abundant lineage in surface oceans conveys a story that will not benefit from the inclusion of additional analyses. "

I don't understand how this statistic can be "a valuable addition", yet the story "will not benefit from the inclusion of additional analyses.". The deep learning analysis is still not intuitively explained in the new version (not the authors' fault, as these methods are designed to be complex). It should be complemented by fixation index calculations, which are simple, and would allow the readers a much greater insight into the basis of the results.

---

## [Author Response]

[Editors’ note: the author responses to the first round of peer review follow.]

Reviewer #1:

This deals with the causes of patterns of diversity in a widespread marine microbe. The data come from a set of contigs assembled from shot gun sequencing of 21 genomes onto which are read-mapped metagenomic data from 103 samples. The focus is a particular clone that is particularly abundant in southern samples. A core genome is assembled from the metagenomic data and this is interrogated for signatures of selection using a curious approach based on physico-chemical properties of amino acids. A further step looks for correlations with water currents.At first glance I found the paper reasonably clear, but the closer I read, the more murky things became. In fact I am not too sure what the authors have really done and I am not persuaded that their main claim, that selection is responsible for biogeography in S-LLPA, is supported by their data. Overall the level of explanation is insufficient to allow the reader to understand what has been done.

We thank the reviewer for their interest in our study. We agree with the reviewer’s criticism regarding the lack of clarity of our initial submission, which made it difficult for readers to follow our analyses and findings. These concerns prompted us to go back to the drawing board and improve the clarity of our manuscript as well as substantiate our claims with new analyses.

Our revised manuscript includes a simplified terminology, an entirely re-written Introduction, and a vastly modified Results section that clarifies both the question we set out to investigate, and the novelty of the approaches we developed to address it. It also includes new analyses to support our main claim. These analyses include (1) the percent identity distribution of recruited short metagenomic reads for each SAR11 genome, (2) the investigation of the relative frequency of amino acids involved in SAAVs, to demonstrate the link between purifying selection and hydrophobicity, and (3) the identification of thousands of SAAV allele frequency trajectories correlating with in situ temperature. One critical point is that we now follow more common naming conventions for SAR11 clades and sub-clades. Thus, in our revision we now define six sublineages within 1a.3 by linking ecology, phylogeny, and gene content of available SAR11 genomes, and replace the term S-LLPA with 1a.3.V.

Overall, these new analyses support our main claim that environmentally-mediated selection is a strong determinant of SAR11 evolution and biogeography by providing more evidence for the linkage between SAAVs and in situ temperature at the highly resolved level of allele frequency trajectories for individual codon positions across metagenomes.

Among the difficulties:Title / Abstract: The title states that biogeography is governed by selection, but in the Abstract the authors use more temperate language. So I guess they are not so sure. The Abstract claims that systematic purifying selection and adaptive mechanisms governing non-synonymous variation have been identified, but at best what they have are signatures consistent with expectations. I disagree that analysis of sequence data reveals different niches (it may reveal the existence of ecotypes that may be indicative of distinct niches). What is a proteotype?

We have replaced “natural selection” by “environmentally-mediated selection” throughout the manuscript to clarify our findings, and incorporated new analyses corroborating the role of selection in the global biogeography of 1a.3.V. We have also changed our title to better reflect our findings.

We define a proteotype as homogeneous configuration of environmental single-amino acid variants in the context of a microbial lineage we access through a reference genome and metagenomic read recruitment. We have defined the six ‘proteotypes’ in our revised manuscript from sets of metagenomes that are clustered together based on environmental single-amino acid variant profiles. Thus, these proteotypes correspond to oceanic regions that shared similar amino acid variations for 1a.3.V. Respecting the controversy surrounding the term, we adopt a simple definition of ‘niche’ as in “the fit of a [population] living under specific environmental conditions” (doi:10.1007/978-94-017-9014-7_26). If proteotypes reflect environmental conditions, which is the main premise of our study, as they reflect oceanic current temperatures, we do not see why the sequence data analyses would not be insufficient to distinguish niches. Hence, we prefer to keep the term in our manuscript, and hope the reviewer agrees with our resolution.

Place of isolation of the focal 21 SAR11 genomes is important in a study of biogeography, but no attention has been given to this.

We agree with the reviewer that it is an intuitive consideration that we should have been more careful in addressing. Our analyses show that the location of isolation (coastal Hawai’i) has very little relevance to the biogeography and population genetics of the genome. 1a.3.V is not particularly more prevalent, or less diverse in samples closer to the location of isolation. We now clarify this point in two sections of the text:

“Strain HIMB83 contains a 1.4 Mbp genome with 1,470 genes, and was isolated from coastal seawaters off Hawai’i, USA. Despite this, it recruited more reads from distant locations than from samples closest to the source of isolation (Supplementary file 1C).”

“The SAAV density (the percentage of codon positions that harbor a SAAV) of core 1a.3.V genes averaged 5.76% and correlated with SNV density (19.3% on average) across the 74 metagenomes (R2=0.89; Figure 1—figure supplement 1). SNV and SAAV density metrics did not decrease in metagenomes sampled closest to the source of isolation, suggesting that the location of isolation for strain HIMB83 does not predict the biogeography and population genetics of 1a.3.V.”

Our revision also includes a new analysis that summarizes percent identity statistics of all metagenomic reads recruited by HIMB83 to substantiate this observation.

There are many challenges in drawing conclusions from metagenomic data via read-mapping to a reference. Particularly problematic is the fact that diversity of closely related organisms within a sample is not known. Thus when many SNPs are detected is this indicative of a diverse population, or is it a signature that comes from a single over-represented clone that is rather divergent? Similarly, the authors take high similarity as signature of a recent sweep, but this might reflect a single over-represented genome from a closely related organism. I'm not sure how the authors dealt with these and similar problems.

We thank the reviewer for their insightful question and an opportunity to clarify this point, which in fact is an important aspect of our study. While we rely on reference genomes for metagenomic read recruitment, our analyses of genetic heterogeneity in the environment through singlenucleotide variants (SNVs) and single-amino acid variants (SAAVs) rely on the consensus of the environment rather than divergence from the reference sequence. In other words, we established our SNVs and SAAVs based on the information content within a metagenome, rather than by calculating distance between a metagenome and a reference genome. Thus, if there was a single over-represented clone in the environment as the reviewer suggests, there would have been no SNVs or SAAVs identified from that metagenome: despite the disagreement between that clone and HIMB83, the environment would reveal subtle variation within the clone. In contrast, we found many variants and remarkable diversity that differed across metagenomes, which allowed us to to place this variation in the context of geography. The new Figure 2 in our revised manuscript substantiates this point.

Sweeps could occur at the genome-level causing over-representation of a single genome (which we did not observe for SAR11 in any of the metagenomes analyzed here) or at the gene-level when all closely related members of a diverse lineage share the same gene (which we also rarely observed). Thanks to SAAVs, we also had the opportunity to study conservancy of genes at the level of amino acid sequences. These analyses also suggested a lack of sweeps, and instead revealed the systematic occurrence of different variants in sets of metagenomes that originated from distant geographic locations. These two observations support the hypothesis that there was not a single over-represented clone in any of the samples analyzed here.

With regard to selective sweeps, the results of the analysis appear not to be given (subsection “SNVs to SAAVs: Accurate characterization of non-synonymous variation”, gives the conditions that need to be fulfilled in order for a protein to have undergone a sweep, but not the results).

Supplementary file 2D describes SAAV density for each gene across the 74 metagenomes; values of zero indicate no variation in the environment. The reviewer’s comment prompted us to be more careful with the terminology we use to describe this particular observation, which is the lack of variation rather than an event of gene or protein sweep, which requires more evidence than what our data can afford. Our Figure 5 now summarizes the percentage of invariant proteins across proteotypes and, to address this particular reviewer concern, we extended the section where invariant proteins are described with the following addition:

“We considered a protein to be ‘invariant’ (i.e., absence of variation due to intensive purifying selection acting on introduced mutations) in a given metagenome if it lacked SAAVs; such invariant proteins were rare in our data. In total, we detected 2,548 invariant proteins (only 4.3% of all possibilities across the 74 metagenomes) that encompassed only 113 genes. In addition, all genes, except one 679 nucleotide long ABC transporter (gene id 1469), contained at least one SAAV in at least one metagenome, revealing a wide range of amino acid sequence diversification among core 1a.3.V proteins (Supplementary file 2D).”

Patterns of amino acid substitution types seems an interesting way to go, but this needs to be unpacked, explained and justified.

We thank the reviewer for their guidance. AASTs indeed effectively summarize the multidimensional information the complex SAAV table carries. We now have significantly edited the related section for clarity:

“Next, we sought to investigate amino acids that co-occur in variable sites. SAAVs were often dominated by a few amino acids; hence, the frequency vector for a given SAAV contained many zero values. […] Expectedly, ‘glycine/tryptophan’ was exceedingly rare in our data and occurred only once in 738,324 SAAVs (Supplementary file 2H).”

In addition, where we investigated the correlation between AASTs and temperature, we described our results the following way, further justifying its relevance:

“Within subclade 1a.3.V the distribution of AAST frequencies was notably constrained across geographies (Figure 3B). […] Yet, close inspection of AAST frequencies revealed that amino acid exchangeabilities subtly diverge in a pattern that correlates with water temperature and/or its co-variables (Figure 3B insets, Figure 3—figure supplement 2).”

Finally, we relied on AASTs to clarify temperature-associated exchange rates between amino acids in AASTs, which further explains their use:

“The results of this analysis also provide an opportunity to study the directionality of exchange rates of AASTs. […] The most statistically significant asymmetry was between the most common AAST, isoleucine and valine, in which valine was highly preferred in warmer waters (binomial test p-value: 5.7×10^-6^) despite chemical structure similarities.”

Partitioning of SAAV between warm and cold currents: there seems to be a correlation with clustering based on SAAVs, but as above, I need to have more explanation. Much relies on the robustness of the SAAV profile clustering (Figure 3A) and there is no indication of its statistical likelihood.

Here we reanalyzed our data using also a more conventional approach rather than Deep Learning analysis to demonstrate that our findings are preserved even with a different clustering strategy. Deep Learning provided us with a distance matrix that estimates the dissimilarity between 74 metagenomes based on their SAAV profiles, and a hierarchical clustering of these data enabled us to identify 6 clusters. To confirm this outcome was not a by-product of our bioinformatics approach, we also performed an analysis of the SAAV table using Euclidian distance and Ward’s linkage. The figures in Author response images 1 and 2 shows the two main clusters, which are identical to the two main clusters Deep Learning revealed.

Asking for 6 clusters also provided us with near identical organization of metagenomes:

**Author response image 2. respfig2:** 

We favored the Deep Learning approach since its unbiased distance estimation resulted in more highly-resolved branches in the resulting dendrogram.

To further link the clustering of metagenomes based on SAAV profiles, and the underlying sequence composition, in our revised Figure 4A we added the percent identity curves of recruited reads at the bottom of the dendrogram. The strong association between percent identity curves and clusters indicate that our SAAV clustering approach was able to infer the signal that underpins the differentiation among these metagenomes in the context of a single lineage:

We hope these additional insights give the reviewer more confidence in the patterns we observe, and adequately demonstrate that these results are not due to an algorithmic anomaly.

The crunch of the paper is rejection of the neutral model, but this is done by simply saying that neutral models could not explain the observed patters of amino acid substitutions, but no evidence is provided. Also, Figure 4, which I hoped would enlighten me, does not exist.

We regret that Figure 4 was not available to the reviewer:

We hope the remarkable differences between the neutral model and the clusters identified through our analysis (most strikingly the group D, C, and E that connect distant geographies) offers the reviewer better insights.

Our new analysis of allele frequency trajectories adds further statistical support to our claim, corroborating the biogeography of proteotypes. We wrote:

“In addition to considering patterns of variability that emerged when we pooled data across 37,416 codon positions exhibiting variation within the core 1a.3.V genes, we also investigated the allele frequency trajectories of individual positions (i.e., the relative frequency between the two most prevalent amino acids across the 74 metagenomes) and sought to identify those that correlate with in situ temperature and/or its co-variables. […] It is therefore most plausible to conclude that allele frequencies at these positions are predominantly shaped by environmentally mediated selection.”

And we now end the Results and Discussion section with:

“The striking connection between geographically distant regions of the oceans through SAAVs render neutral processes an unlikely candidate for evolutionary processes that maintain the genomic heterogeneity within 1a.3.V (Figure 4—figure supplement 5). […] Overall, these results indicate that environmentally-mediated selection is a strong determinant of SAR11 evolution and biogeography.”

In general I found the figures overly dense and very difficult to understand given the paucity of explanation in the captions. Take Figure 1: latitudinal gradient, I presume north is at the top? Colouring seems inconsistent and so on.

We thank the reviewer for their attention. We now have clarified the latitudinal gradient in Figure 1 (the reviewer was correct in their guess: the organization of metagenomes top to bottom followed North to South). We have also tried to strategically use color to enhance the clarity of new figures incorporated into the resubmission.

We agree with the reviewer that some of the figures are dense with information. We have dedicated considerable time to create them so that they could convey relevant information from our holistic analyses (e.g., the SAR11 metapangenome in Figure 1A) and the new concepts introduced in this study (e.g., SAAVs in the context of inferred protein structures in Figures 3C and 4A). Carefully selected main figures cover significant aspects of the study, and altogether support our main claim that environmentally-mediated selection plays a role in governing the global biogeography of SAR11.

We thank the reviewer for their efforts to evaluate our study and for their helpful insights.

Reviewer #2:

It's hard to tell who this paper is written for. The bioinformatics applications are the most interesting: identifying the core genome of a lineage of interest from metagenomics data, using a novel algorithm for (somehow) identifying significant subclades within the lineage, and finding a surprisingly high frequency of certain amino acid substitutions that are not predicted by Blosum. There are also interesting implications for ecological diversification, but this aspect of the paper is written without enough explanation or interpretation of data, as I will detail. Also, I think most microbial ecologists would be amazed to see a paper on diversification within "a single population" that is actually not a population at all, but rather a fair chunk of a genus.It is so inappropriate to call the S-LLPA lineage a "single population," since it has nearly 20% sequence divergence within it, and because there is clearly profound ecological diversification within this group. (Given that the Eren group has made tremendous progress toward discovering newly divergent, ecologically distinct populations of bacteria with their MED algorithm, it seems ironic that the group is now calling this phylogenetically huge group a single population.) Calling this group a population seems to have led to the conclusion that "their broad geographic prevalence suggest<s> dispersion is not a limiting factor for SAR11 in surface seawater…." But this conclusion is based on a false reification of all the various true populations within what the authors are calling a population.

We are very thankful for the reviewer’s criticism. In fact, their comments and concerns motivated us to perform new set of comprehensive analyses that helped us improve our intellectual and technical approaches and led to significant improvements in our manuscript. We agree with the reviewer the occurrence of multiple sequence-discrete populations would have influenced our conclusions dramatically, although our new analyses suggest that it is not the case. The new Figure 2 provides new insights into the percent identity of reads core 1a.3.V genes recruited across the 74 metagenomes. These results show that the data suggest the absence of various true populations within the niche of 1a.3.V, and much more clearly reveal the complex nature of this lineage, which was also observed in other studies that investigated sequence-discrete SAR11 populations (we cited multiple of these studies in our new introduction).

We agree with the reviewer that the use of the term “population” to describe 1a.3.V without extensive justification was an oversight on our part. However, we respectfully disagree with the reviewer’s suggestion that the metagenomic reads we competitively recruit through the HIMB83 genome represent a fair chunk of a genus. There is no consensus among microbiologists on what the appropriate theoretical definition of a population should be. While there are considerably wellsupported computational attempts to offer operational definitions for practical conveniences, SAR11 is a well-known and previously documented outlier of these operational definitions (our new introduction makes a more appropriate attempt to cover the literature). Where does a genus start and end? In the absence of an appropriate theoretical definition for a population, the boundaries between different levels of taxonomy are subject to interpretation. We believe, however, that most microbiologists and microbial ecologists would agree that a genus describing multiple ‘species’ must contain multiple sequence-discrete populations (SDPs). In other words, it is conceivable to expect that if metagenomic reads we recruit from the environment using HIMB83 represent a fair chunk of a genus, then there should be multiple SDPs that can be inferred from their levels of identity. However, what we found in our data was a lack of any evidence for SDPs. In fact, the recruited reads showed a much broader identity distribution pattern, similar to what is proposed for SAR11 by leading scientists in the field. Thus, our new short-read-level analyses do not support the hypothesis of the presence of multiple SDPs in our dataset. But ultimately, after much consideration we elected to not use the controversial term ‘population’. Our data gives access to a large number of environmental cells that differ from each other rather dramatically, but not randomly: in a cloud of sequences that can’t be distilled into multiple SDPs, disagreements across the genomes that make up this cloud emerge from very specific locations of genes without the presence of individually abundant clones. Our study distills that information to show that amino acid sequence-level diversification connects geographically distinct stations, and we accomplish this by focusing on a prevalent and abundant subclade of SAR11 that has a single cultivated representative.

We have extensively modified our text to better introduce and describe our work. We thank the reviewer for their insights, and hope that our new framework satisfies the reviewer.

The authors talk about S-LLPA offers "a unique opportunity to study the genomic diversity and evolutionary genomics of a single marine microbial population…." Not so unique when the "population" is a huge chunk of a genus.

We have removed this sentence as a part of the changes described above.

The authors do justify their calling this clade a population because it constitutes what is recruited by a single isolate from various metagenomes. But their own work clearly shows, to my mind, that this is not a good way to discover populations.

Based on the reviewer’s comments we have modified the wording accordingly. HIMB83 provided the opportunity to study an abundant and widespread SAR11 lineage (1a.3.V) that genomeresolved metagenomics has thus far failed to provide access. We do agree that read recruitment to isolate genomes can in principle lead to mistakes due to non-specific recruitment; however, we have taken care to guard against this error and show that our approach offers valuable insights when sufficient samples of a population’s genomes are unavailable. Our gene-level recruitment results and the reliance on the core genes of this lineage minimize shortcomings of the use of isolate genomes to support our claims.

The authors have used their Deep Learning algorithm for demarcating six phylogenetic groups, which they call proteotypes, from their S-LLPA lineage. It is frustrating that the authors neglected to give any rationale for their approach. Is it based on sequence clustering of shared genes? Is it based on genome content sharing? Or both? (Even though the algorithm has been published, they should say something briefly about what the inputs are and how it works.)

This important point requires clarification. We first identified 738,324 SAAVs across the 799 core 1a.3.V genes and 74 metagenomes. We then simplified our data by taking into account two most frequent amino acids in each SAAV. To explain our reasoning we have added the following description to the text:

“SAAVs were often dominated by a few amino acids; hence, the frequency vector for a given SAAV contained many zero values. To reduce sparsity, we first simplified our data by associating each SAAV with an amino acid substitution type (AAST), defined as the two most frequent amino acids in a given SAAV.”

We have also clarified the data used by the Deep Learning algorithm (we regret the poor description in our initial submission) and further explained the Deep Learning approach. The relevant section now reads:

“We then sought to extend the concept of allele frequency tracking at individual SAAV positions to investigate the potential for large-scale geographic partitioning within the metagenome dataset. […] Hierarchical clustering of samples based on Deep Learning-estimated distances (Supplementary file 4A) resulted in two main groups: the Western (warm) and Eastern (cold) boundary currents (Figure 4A).”

On two occasions the authors write about protein sweeps. They state that the sweeps are "rare." I think most evolutionary ecologists of bacteria would want to know how rare, and in which genes. Also, it would be important to indicate whether the sweeps traveled across ecologically distinct groups. In the first paragraph of the Discussion, the authors mention that there are more sweeps in warm currents, but this should be fleshed out and made clearer.

We agree with the reviewer that protein sweeps and their linkage to ecologically distinct groups and functions are of particular interest. We have modified the relevant section for clarity and included in our revision a new supplementary table that describes additional statistics:

“We considered a protein to be ‘invariant’ (i.e., absence of variation due to intensive purifying selection acting on introduced mutations) in a given metagenome if it lacked SAAVs; such invariant proteins were rare in our data. In total, we detected 2,548 invariant proteins (only 4.3% of all possibilities across the 74 metagenomes) that encompassed only 113 genes. In addition, all genes, except one 679 nucleotide long ABC transporter (gene id 1469), contained at least one SAAV in at least one metagenome, revealing a wide range of amino acid sequence diversification among core 1a.3.V proteins (Supplementary file 2D).”

While we are happy to see the reviewer finds this of interest, we believe protein sweeps do not represent significant aspects of our study. Yet the data we have made available provide access to the occurrence of invariant proteins in the context of metagenomes (Supplementary file 2D) along with extensive information on metagenomes (Supplementary file 1A) as well as each gene (Supplementary file 1J). We hope the reviewer agrees with our allocation of the limited manuscript space.

Subsection “Purifying selection governs the identity of amino acid substitution types”, first paragraph. It wasn't clear to me whether the authors were claiming that their focus group was unique in having a stable proportion of amino acids on a global scale, or whether other groups are showing the same pattern.

We have revised the section in question to improve its clarity using our new analyses:

“Within subclade 1a.3.V the distribution of AAST frequencies was notably constrained across geographies (Figure 3B). For example, the relative standard deviation of ‘aspartic/glutamic acid’ frequencies across the 74 metagenomes was just 3.0%, and the statistical spread of other AASTs was comparable (Figure 3B). This suggests that there exists some evolutionary mechanism(s) maintaining amino acid exchangeability within subclade 1a.3.V throughout the global surface ocean.”

Our study does not intend to make a claim regarding how unique observed patterns are to microbial taxa. However, we find the question posed by the reviewer an interesting one.

In the subsection “Purifying selection governs the identity of amino acid substitution types”, I don't see how the authors are finding evidence of purifying selection. Rather, it seems they are finding the constraints on adaptive substitutions of amino acids. That's different from purifying selection, which is usually the evolutionary stability of a particular sequence.

We thank the reviewer for bringing up this point. Our revised study includes significant changes to clarify our observations.

Purifying selection, the selective removal of alleles that are deleterious, indeed does not impact permissible amino acid substitutions, but we found evidence that purifying selection shapes the relative frequency of amino acids involved in SAAVs as a function of hydrophobicity, which is supported by the existing literature:

“Hydrophobic interactions within the solvent inaccessible core of proteins are known to be critical for maintaining the stability required for folding and activity, which enforces a strong purifying selection placed on mutations occurring in buried (solvent inaccessible) positions (Bustamante, Townsend and Hartl, 2000; Chen and Zhou, 2005; Worth, Gong and Blundell, 2009). […]Overall, our compositional analysis revealed that the occurrence of amino acids in SAAVs is roughly correlated with the occurrence of amino acids within the core 1a.3.V genes, and that deviations from this expectation are driven in part by levels of purifying selection that depend upon the suitability of an amino acid’s hydrophobicity for a given physicochemical environment (Figure 3—figure supplement 1).”

Besides the relative proportion of amino acids involved in SAAVs, our analysis of inferred 3D structures for hundreds of core 1a.3.V genes provides additional insights into the apparent effect of purifying selection in the occurrence of SAAVs as a function of solvent accessibility:

“Within the subset of the 1a.3.V proteome accessible to us, we found that buried amino acids (0-10% relative solvent accessibility) were approximately 4.4 times less likely to be variant than those that were exposed (41-100% relative solvent accessibility) (ANOVA, pvalue: <2×10^-16^). […] The local physicochemical environment therefore shapes variation, and visual inspection of Figure 3C indicates that this is conserved across distant geographies; i.e. positions that vary in one metagenome are likely to vary in others, as well.”

Strong biases associated with hydrophobicity and solvent accessibility both expose a predominant role of purifying selection in observed amino acid frequencies in SAAVs within the members of SAR11 lineage 1a.3.V. We hope the reviewer appreciates our clarifications.

It's interesting that there are two clades within S-LLPA that are associated with different temperature regimes. And it's interesting that the subclades within these clades are different in the geographic regions where they are most abundantwith similar temperatures. The interpretation of this result depends on whether different cold-adapted subclades are never found together (meaning that they may have no adaptation differences) or whether they sympatric to some extent (meaning that they are coexisting based on ecological differences other than temperature partitioning). It seems that the latter is the case, but the authors should have made this clearer. For example, pie charts showing the relative abundances of the subclades in each region would be useful.

The reviewer brings up a very important point, which was of significant interest to us as well. However, our short-read level analyses suggested that an increased fitness to a given temperature regime does not yield sequence-discrete populations that are distinguishable, which prevents us from making quantitative estimates of their differential abundance across geography. Deconvoluting haplotypes found in metagenomic read recruitment results is possible thanks to the increasing availability of tools (i.e., Lineage, DESMAN, or ConStrains), however, none of them are suitable to resolve the extensive diversity we observe in SAR11 subclade 1a.3.V. We are looking forward to new contributions from single-cell genomics and cultivation to address this significant question our study could not contribute.

Reviewer #3:

The work by Delmont et al. describes a large-scale and thorough analysis of an important marine bacterium population, SAR111. They use multiple approaches to identify signals of selection to understand the evolution and ecology of these microbes. The authors deserve credit for their very clear presentation of data and methods. However, I have multiple concerns regarding the analysis and interpretation of the data:1) The nature, and strength of, selection pressures acting on this bacterial population are not clearly defined.

We agree with the reviewer. The manuscript now better describes the nature and strength of selection pressures acting on 1a.3.V. Especially, we show that (1) hydrophobicity and solvent accessibility influences the strength of purifying selection acting on amino acids, and (2) few AASTs along with thousands of allele frequency trajectories significantly correlate with in situ temperatures, in line with the biogeography of proteotypes. Our response below clarifies these points further.

a) The description of the substitution analysis "suggests a powerful influence of purifying selection", while also "suggesting a role for adaptive processes for functional diversification of S-LLPA proteins". Which of these dominates? Could this just be noisy neutral evolution – what is the expected variation of the S statistic, and the SAAV patterns, under a neutral scenario?

This is an important consideration and requires a comprehensive clarification. Our revision no longer contains the sentences the reviewer cited, and better clarifies that our investigation of purifying selection acting on 1a.3.V is independent of the debate regarding neutral evolution versus natural selection. Simply, we observed that certain amino acids (mostly hydrophilic), and most notably, few AASTs (e.g., alanine/isoleucine) predominated in our SAAV table.

Regarding the amino acids, we now write:

“Hydrophobic interactions within the solvent inaccessible core of proteins are known to be critical for maintaining the stability required for folding and activity, which enforces a strong purifying selection placed on mutations occurring in buried (solvent inaccessible) positions (Bustamante, Townsend and Hartl, 2000; Chen and Zhou, 2005; Worth, Gong and Blundell, 2009). […] Overall, our compositional analysis revealed that the occurrence of amino acids in SAAVs is roughly correlated with the occurrence of amino acids within the core 1a.3.V genes, and that deviations from this expectation are driven in part by levels of purifying selection that depend upon the suitability of an amino acid’s hydrophobicity for a given physicochemical environment (Figure 3—figure supplement 1).”

Besides the relative proportion of amino acids involved in SAAVs, the analysis of inferred 3D structures for hundreds of core 1a.3.V genes provides additional insights into the apparent effect of purifying selection on the occurrence of SAAVs (role of solvent accessibility this time):

“Within the subset of the 1a.3.V proteome accessible to us, we found that buried amino acids (0-10% relative solvent accessibility) were approximately 4.4 times less likely to be variant than those that were exposed (41-100% relative solvent accessibility) (ANOVA, pvalue: <2×〖10〗^(-16)). […] The local physicochemical environment therefore shapes variation, and visual inspection of Figure 3C indicates that this is conserved across distant geographies; i.e. positions that vary in one metagenome are likely to vary in others, as well.”

Regarding the AASTs, we found that predominant ones corresponded to competing amino acids with similar properties, shedding a different light on purifying selection. We described our findings in our revision:

“In 738,324 SAAVs, we observed 182 of 210 theoretically possible unique AASTs at varying frequencies (Supplementary file 2H). […] Expectedly, ‘glycine/tryptophan’ was exceedingly rare in our data and occurred only once in 738,324 SAAVs (Supplementary file 2H).”

These points are interesting, but they do not favor natural selection over neutral evolution. Instead they provide information regarding permissible versus non-permissible diversification. On the other hand, analysis of the permissible SAAVs can shed light in the natural selection versus neutral evolution debate. This is now covered in subsequent sections of our revision to help the reader. For instance, we found that some AASTs, and thousands of allele frequency trajectories, significantly correlated with in situ temperature, suggesting an important role for natural selection. We first wrote:

“Within subclade 1a.3.V the distribution of AAST frequencies was notably constrained across geographies (Figure 3B). […] Yet, close inspection of AAST frequencies revealed that amino acid exchangeabilities subtly diverge in a pattern that correlates with water temperature and/or its co-variables (Figure 3B insets, Figure 3—figure supplement 2).”

We then found that for some AASTs, one amino acid tended to be avoured by temperature increases compared to its competing amino acid genome-wide. We added:

“The results of this analysis also provide an opportunity to study the directionality of exchange rates of AASTs. […] The most statistically significant asymmetry was between the most common AAST, isoleucine and valine, in which valine was highly preferred in warmer waters (binomial test p-value: 5.7×10^-6^) despite chemical structure similarities.”

Finally, we determined that thousands of 1a.3.V allele frequency trajectories correlated with in situ temperature, in line with the biogeography of proteotypes. We concluded:

“In addition to considering patterns of variability that emerged when we pooled data across 37,416 codon positions exhibiting variation within the core 1a.3.V genes, we also investigated the allele frequency trajectories of individual positions (i.e., the relative frequency between the two most prevalent amino acids across the 74 metagenomes) and sought to identify those that correlate with in situ temperature and/or its co-variables. In 2,740 of the 37,416 positions, the null hypothesis that no correlation with temperature exists was rejected (Supplementary file 3B; p-value < 0.01). Figure 3—figure supplement 3 illustrates example cases and correlation statistics per AAST. It is statistically implausible that such correlations with temperature could have arisen from neutral evolution, given that distant oceans share similar temperatures (Supplementary file 1A). It is therefore most plausible to conclude that allele frequencies at these positions are predominantly shaped by environmentally mediated selection.”

Regarding the last question of the reviewer: under a neutral scenario, SAAV patterns should have reproduced trends depicted in Panel C of this supplemental figure (patterns we observed are different, and favor natural selection due to the distant metagenomes sharing similar SAAV profiles, for which Proteotype D offers a remarkable demonstration), see Figure 4—figure supplement 5.

b) The neutral model of Hellweger et al. still predicts geographic separation of strains, which should be made clear in the Introduction. A falsification of this would be a repeated association of bacterial strains with ecological niche. This appears to be the case at the highest level of the hierarachy inferred with "deep learning", but at lower levels, predictors are genetic, not ecological, and therefore likely represent characteristics that were identified by the clustering algorithm itself. This might be expected under a neutral model.

In principle, the reviewer is correct. But we did not observe variation at the level of SAAVs that fit the neutral model, even at finer scales of evolutionary distance. At the level of two main groups (Western versus Eastern boundary currents) as well as at the level of individual proteotypes, the clustering of metagenomes based on differential occurrence of SAAVs favors natural selection (now introduced as environmentally-mediated selection). Proteotype D is a particularly good example connecting the Red Sea and two sides of the Panama despite considerable distances.

Proteotype C connects remarkably distant, yet polar regions of the ocean. Similarly, Proteotype E also connects distant locations. Neutral evolution alone could lead to such patterns, as indicated by the simulation (Figure 4—figure supplement 3, panel C).

To cover this point, the Introduction now reads:

“At the level of individual populations, a key simulation by Hellweger et al., 2014, showed that the intra-population sequence divergence that reflects the geographic patterns of distribution for SAR11 cells could emerge solely as a function of ocean currents, without selection (Hellweger et al., 2014).

And the Results and Discussion reads:

“The striking connection between geographically distant regions of the oceans through SAAVs renders neutral processes an unlikely candidate for evolutionary processes that maintain the genomic heterogeneity within 1a.3.V (Figure 4—figure supplement 5). In fact, both the main ecological niches and more refined proteotypes indicate that SAAVs are not primarily structured by the global dispersal of water masses but instead tend to link distant geographic regions with similar environmental conditions (Figure 4B). This is corroborated by thousands of allele frequency trajectories correlated with in situ temperatures. Overall, these results indicate that environmentally-mediated selection is a strong determinant of SAR11 evolution and biogeography.”

2) The authors have opted against traditional population genetics methods. It is possible that the new methods are a substantial improvement, but they need to be justified. Some examples:a) The authors justify using the "S" statistic rather than d(N)/d(S) based on there being many instances of multiple substitutions within the same codon. But this is exactly what d(N)/d(S) is meant to correct for, and S does not capture – there may be many synonymous substitutions on an evolutionary pathway, but if only one of them is non-synonymous, then S will only count this only as a non-silent change, underestimating purifying selection. What is the neutral expectation for S?

The “S” statistic (introduced as SAAVs/SNVs ratio) is not central to our study but provides an opportunity to link SNV and SAAV densities at the level of individual genes or entire metagenomes. We observed interesting trends, such as the increase of this ratio in cold waters (proteotypes A, B and C).

The accuracy of d(N)/d(S) deteriorates if a given codon contains more than a single SNV (the URL http://merenlab.org/2015/07/20/analyzing-variability/#single-codon-variants also contains some visual support for this statement with a mock example). A substantial fraction of codons analyzed in our study contained more than one SNV, a critical aspect of our study that is now better introduced in our revised manuscript. In those situations, SNVs need to be analyzed simultaneously, since the effect of one SNV (whether it is synonymous or not) will depend on the occurrence of others. As far as we know, there is no published study that effectively takes into account multiple SNVs per codon without calculating the exact codon frequencies to estimate synonymity accurately. Since they rely on actual codon frequencies and not pileups of nucleotide positions that are independent of each other, we hope the reviewer agrees that SAAVs will provide a better metric of synonymity compared to d(N)/d(S) in the case of lineages with extremely high SNV densities.

Under the neutral model, changes in the SAAVs/SNVs ratio must tend to associate with geographical adjacency as a function of the movement of water mases, rather than environmental parameters. The significant shift in SAAVs/SNVs ratio between cold currents and warm currents is of relevance to our study, especially in light of other insights (especially, the linkage between allele frequency trajectories and in situ temperature, and the biogeography of proteotypes), favors the predominant role of natural selection acting on the permissible amino acid diversification traits of this global SAR11 lineage.

Overall, we do not suggest that the SAAVs/SNVs ratio should replace d(N)/d(S). Simply, SNV density was high for 1a.3.V and many SNVs co-occurred in the same codon, rendering classical d(N)/d(S) analyses limited, which had never been challenged with extremely complex environments, if not completely irrelevant.

We appreciate reviewer’s input as it helped us better clarify the relevance of SAAVs regarding synonymy in our revision in multiple places. Specifically, the Introduction now reads:

“The substantial sequence diversity within environmental SAR11 populations not only explains the absence of SAR11 population genomes in genome-resolved metagenomics studies, but also challenges conventional approaches to the study of population genetics in microorganisms. For instance, the multiple occurrence of single-nucleotide variants in individual codon positions would render commonly used computational strategies that classify synonymous and non-synonymous variations based on independent nucleotide sites (such as in (Schloissnig et al., 2012; Bendall et al., 2016)) unfeasible.”

And the revised Results and Discussion reads:

“Percent identity distributions are useful to assess overall alignment statistics of short reads to a reference; however, they do not convey information regarding allele frequencies, their functional significance, or association with biogeography. […] Thus, quantifying frequencies of full codon sequences is a requirement to correctly assess synonymity, and it is this principle we employed in our workflow to extract SAAVs.”

And finally, regarding the specific case of the SAAVs/SNVs ratio, we added the following section:

“Interestingly, the niche defined by cold currents exhibited significantly more SAAVs (ANOVA, p-value: 1.66×〖10〗^(-12)) and a significantly higher SAAVs/SNVs ratio (ANOVA, p-value: 1.07×〖10〗^(-10)). This observation could be explained either by (1) extinction/re-emergence events that operate continually on specific codon positions (adaptive evolution), or (2) changes in abundances within a large seed bank of variants due to positive and negative selection as the lineage transits.”

We hope the reviewer agrees with our changes.

b) The BLOSUM matrices are used as "as a proxy to assess the functional difference between the pairs of amino acids", which are then compared to the substitution frequencies – but this is circular reasoning, as these matrices are calculated from substitution frequencies. The versions used in the paper are also inappropriate for this dataset, as they were calculated using more closely-related sequences than are compared in this paper.

We agree with the reviewer. We have removed BLOSUM matrices from our revision.

c) "Deep learning" is applied to cluster populations together through a method designed without genetic data in mind. This does not mean it is wrong, but as it is not an intuitive approach, it would be helpful to see validation against a more standard method, such as the fixation index, F(ST). This would help understand the distinct properties of the six subclusters – for instances, proteotype D looks to be a single successful "strain" – is this expected to be picked up as a signal by the "deep learning"?

First, we would like to address reviewer’s suggestion regarding the nature of Proteotype D. Our analysis of SAAVs using Deep Learning estimated distances between 74 metagenomes based on single amino acid variants we observed in them in the context of a single genome. This analysis linked 74 metagenomes to six proteotypes. Proteotype D emerged as a tight cluster despite its geographical spread, as the metagenomes it represents displayed similar SAAVs. If this was a single successful strain, we would have expected to find zero divergence from the environmental consensus for SAAVs. However, this was not the case: although much less when compared to other proteotypes, proteotype D did contain >5,000 SAAVs per metagenome (Figure 5, panel A). In addition, our new analysis of the percent identity of recruited reads also showed that proteotype D still maintains a large degree of variability and is far from a state that can be considered nearclonal.

Regarding the fixation index, we agree with the reviewer that this would have been a valuable addition to our work. However, we respectfully disagree with the reviewer regarding the level of intuitiveness of our approach, and we hope that our revision did a better job at introducing its relevance to study this lineage. Besides, the application of classical population genetics methods to these data are challenging due to technical reasons we outlined in our response regarding the remarkable complexity of these populations. We are hoping that our study will cultivate interest from the experts of population genetics to develop new methods that can deal with intra-population diversity of complex environmental agglomerates through ‘omics data. Our study formally introduces strategies and analytical tools for (1) the concept of SAAVs and AASTs, (2) the linkage between environmental variants and predicted protein structures, and (3) the linkage between allele frequency trajectories and environmental variables. We believe the significant amount of methodological development in this work that offers descriptions of the biogeography of the most abundant lineage in surface oceans conveys a story that will not benefit from the inclusion of additional analyses.

On the reviewer concern regarding the validation of Deep Learning: as we mentioned in our response to the Reviewer #1, we did use a more standard approach to cluster metagenomes based on the same SAAV frequency table as well. Here we would like to share parts of our previous response with the Reviewer #3 in case they do not have access to other responses: To make sure our findings with Deep Learning were not a by-product of our bioinformatics approach, we also performed an analysis of the SAAV table using Euclidian distance and Ward’s linkage. The figure in Author response images 1 and 2 shows the two main clusters, which are identical to the two main clusters Deep Learning revealed

Despite these results we elected to use the Deep Learning approach since the unbiased distance estimation afforded by this approach resulted in more highly-resolved branches in the resulting dendrogram.

3) Quality of read mapping. Many of the sequences being compared are highly divergent, but most mapping software is intended to align reads to highly similar reference sequences. While there is a promising correlation between the SNP densities identified by BWA and BOWTIE2, there is a greater than two-fold difference in the absolute densities. This suggests a very high false positive or false negative rate, depending on the method. The authors need to validate the performance of these algorithms when using highly divergent reads. Additionally, what is the justification for using the "the base-5 logarithm of the mean coverage of a gene remained within {plus minus}1 of that for the mean coverage across all genes" for assessing consistency of coverage?

We agree with the reviewer that mapping stringency is indeed critical to study population genetics, especially when one relies on reference genomes and recruitment of metagenomic reads. In most microbial lineages, the extent of diversity within a population is rather low and stringent mapping stringencies (e.g., >95% sequence identity) can be applied. This, however, is not the case for SAR11. To better clarify this point, we have significantly modified the Introduction. The relevant section now reads:

“Interestingly, the boundaries of environmental SAR11 populations appear to not comply with the 95% ANIr cutoff. For instance, Tsementzi et al., 2016, observed substantial sequence diversity within sequence-discrete SAR11 subclades in the environment, and suggested that an ANIr as low as 92% would be required to adequately define the boundaries of the SAR11 populations recovered in their study (Tsementzi et al., 2016). These findings are consistent with a comprehensive study of isolate genomes and marine metagenomes by Nayfach et al., 2016, which suggested that SAR11 is one of the most genetically heterogeneous marine microbial clades (Nayfach et al., 2016). The substantial sequence diversity within environmental SAR11 populations not only explains the absence of SAR11 population genomes in genome-resolved metagenomics studies, but also challenges conventional approaches to the study of population genetics in microorganisms.”

In line with observations from Tsementzi et al., 2016, our new analyses of short reads recruited by HIMB83 revealed extensive diversity, suggesting lower mapping stringencies would benefit deeper insights into SAR11 population genetics. During our in-house tests we found that using a stringency cut-off of 95% resulted in drastic changes in coverage values for Ia.3.V, and led decreased stability of coverage across metagenomes which resulted in much less number of core genes and metagenomes to work with. After exploring different software mapping algorithms and parameters, we concluded that the decreased SNV density with BWA was most likely due to false negatives (gene sections with high degree of SNVs were entirely overlooked). Using Bowtie2 allowed us to study the heterogeneity within this lineage at the amino acid variants level using DNA sequences that were up to 12% divergent from the reference (Figure 2—figure supplement 1 displays similar trends for all other isolate genomes).

Regarding the “base-5 logarithm”: we thank the reviewer for pointing it out. We have now better clarified our strategy:

“We then defined a subset of HIMB83 genes as the core 1a.3.V genes if they occurred in all 74 metagenomes and their mean coverage in each metagenome remained within a factor of 5 of the mean coverage of all HIMB83 genes in the same metagenome. This criterion accounted for biological characteristics influencing coverage values in metagenomic surveys of the surface ocean such as cell division rates and variations in coverage as a function of changes in GC-content throughout the genomic context. Figure 1—figure supplement 1 displays the coverage of all HIMB83 genes across all metagenomes, and Supplementary file 1J reports the coverage statistics.”

We would also like to acknowledge that given the complexity of both SAR11 lineages and metagenomics, it is indeed difficult to defend any heuristic to identify core genes. We fully appreciate this limitation, and to address further concerns we made an attempt to better explain the relevance of the 799 core 1a.3.V genes we identified:

“While the 799 genes that met these criteria systematically occurred within the niche boundaries of 1a.3.V, 40% of the remaining 671 HIMB83 genes that were filtered out were present in five or less metagenomes and coincided with hypervariable genomic loci (Figure 1—figure supplement 1). Hypervariable genome regions are common features of surface ocean microbes (Coleman et al., 2006; Zaremba-Niedzwiedzka et al., 2013; Kashtan et al., 2014; Delmont and Eren, 2018) that are not readily addressed through metagenomic read recruitment but do influence pangenomic trends. Here, less than 10% of gene clusters unique to HIMB83 were among core 1a.3.V genes (Figure 1A), indicating HIMB83’s unique genes are mostly accessory to the 1a.3.V lineage. In contrast, more than 80% of gene clusters that were core to the 21 SAR11 genomes matched to the core 1a.3.V genes. The overlap between environmental core genes of 1a.3.V revealed by the metagenomic read recruitment and the genomic core of SAR11 revealed by the pangenomic analysis of isolate genomes suggests that these genes represent a large fraction of the 1a.3.V genomic backbone (Figure 1A).”

We hope the reviewer agrees with our resolution.

[Editors' note: the author responses to the re-review follow.]Collected reviewer comments:The paper is vastly improved since the original version. It is now much clearer and more compelling, and the paper deals with the issue of populations better.The paper is fascinating for showing that whatever cohesion occurs at >95% ANI in pretty much every other bacterial group does not occur here in SAR11. The paper is also really interesting for giving a robust and sensitive way to identify ecologically distinct metagenome groups from sequence data.My concerns are mostly focused on what the authors have found out about ecological diversity within the 1a.3.V lineage.First, I appreciate that the authors are trying to be more careful about using the concept of population, but there are still places where they are still unclear about what they mean. The authors write of "remarkable intra-population genomic diversity…." It's not clear exactly what the authors mean by "population" here, but in this context what they mean is very important. If there isn't a good definition of population being used, then how can one talk about one population's intra-population genomic diversity being higher than another? It all depends on what you mean by population, and that's not clear here.

The reviewer comment made us realize that our attempt to tip-toe around this issue by trying to avoid the use of the term ‘population’ did not improve the clarity of our study. Instead, it became more confusing due to uncommon and imprecise use of other terms (such as ‘lineage’) to describe what we are studying. We also came to the realization that most studies that use the term ‘population’ do not offer an explicit description (i.e., doi:10.1126/science.1248575), adding more to the confusion. We have now updated the manuscript with the following paragraph, which includes an explicit operational definition to clarify what we mean by a ‘population’ from a practical perspective with its shortcomings and cites insightful studies that offer deeper theoretical and practical discussions for this historical challenge:

“Overall, our data confirm that ANIr values of >95% used previously to delineate sequence-discrete populations does not apply to SAR11. […] However, due to the incomplete theoretical foundation and limitations associated with the use of short metagenomic reads, in discussions here we more conservatively assume that our reads originate from multiple closely related yet intertwined SAR11 populations within subclade 1a.3.V.”

We thank the reviewer for their persistence on this matter.

Figure 2 purports to show that there are not sequence-discrete populations within 1a.3.v, at least not in the sense that Kostas Konstantinidis demonstrates them in his studies, most recently in Jain et al., 2018. It is really interesting that all the SAR11 clades (with just this one clade shown in Figure 2) do not show the drop-off in recruitment of metagenome sequences below about 95% ANI. The authors argue that part of the great sequence diversity of what is recruited into this lineage (that is with much lower ANI) emerges from there being ecological diversity within the lineage (adaptations to different temperatures among different sublineages). I don't think this is a reasonable explanation, as almost every species taxon, nearly all obeying the 95% ANI drop-off rule, is also ecologically heterogeneous. I have suggested, in an interpretation of the recent Jain et al. paper, that the cohesion holding a species taxon together stems from an ability of recombination to act frequently at >95% ANI, plus the possibility of adaptive genes passing between ecologically distinct populations that are >95% ANI (Cohan, 2019) I'd argue that there is more potential for either recurrent recombination to limit divergence among ecologically distinct populations in SAR11, or greater opportunities for a single gene to be adaptive across different populations.

Very helpful insights. We believe the reviewer will find in our revision additional analyses that more clearly demonstrate that the temperature-driven changes in variable positions constitute a very small fraction of variable amino acids. We also offer a clearer discussion of populations in general and alternative interpretations of sequence heterogeneity within 1a.3.V. As a reminder, we also reworded a related paragraph to highlight possible forces driving cohesiveness within a species:

“Both high recombination rates between cells displaying low gANI values and frequent transfer of adaptive genes between ecologically distinct clades could explain the high-level of cohesion between SAR11 populations in the surface ocean (Vergin et al., 2007; Cohan, 2019). […] This analysis revealed a significant correlation between in situ temperature and distribution shape (mean p-value: 2.0×10^(-3); standard deviation p-value: 3.4×10^(-8); skewness p-value: 1.0×10^(-8)), which suggests a strong influence of temperature and its co-variables on the sequence heterogeneity within 1a.3.V (Figure 2) and is incompatible with the hypothesis of random sequence variants.”

I'd like to add that whatever cohesion occurs at >95% ANI in most species taxa does not address the ecological diversity within species taxa, or whether there are sequence-discrete populations within each taxon. In fact, most species taxa have sequence-discrete and sequence-discoverable ecologically distinct populations. (This is the point of the second half of my Current Biology dispatch I mentioned.) So, I don't think that Figure 2 makes an argument that there are not sequence-discrete, ecologically distinct populations within the focus lineage.

We agree with the reviewer. ANI >95% is nothing but an empirical observation that delineates species boundaries and does not suggest anything regarding the within-species population structures. But various values of ANI that are above 95% are often used by others as a practical expedient to define sequence-discrete populations in metagenomic read-recruitment studies (i.e., doi:10.1038/ismej.2015.241, doi: 10.1371/journal.pbio.0060177, etc), and these practical cutoffs slowly become *de facto* expectations. The data in Figure 2 shows that if we were to follow a similar strategy for SAR11 by selecting a sequence similarity cutoff of ANI >95%, we would have been arbitrarily cutting a biologically relevant continuum. We are (and the entirety of our study is) in agreement with the reviewer on this point; the purpose of the sentence the reviewer quotes from our study is to clarify the inappropriateness to apply the commonly used cutoffs to our data.

The authors have noted that there is a small minority of genes that are invariant at the amino acid level across a given metagenome. In the earlier version, they interpreted these observations as evidence of sweeps, but here they have chosen not to address the dynamics that can be inferred by these invariant genes. I think that such single-gene sweeps are very interesting, and have been the topic of discussion of some very interesting papers, including papers by Jesse Shapiro and by Rex Malmstrom. I'd encourage the authors to bring this discussion back. I'll mention that these instances provide evidence that a generally adaptive gene has passed by recombination across all the ecologically distinct populations within a lineage, but I wouldn't expect the authors to necessarily buy in to that interpretation.

We removed the section on ‘protein sweeps’ because we realized that in the absence of timeseries data it was not straightforward to suggest invariant proteins were evidence of protein sweeps or intensive purifying selection. We changed our text to remind the reader both alternatives by adding this sentence: “We considered a protein to be ‘invariant’ (i.e., absence of variation due to intensive purifying or positive selection) in a given metagenome if it lacked SAAVs”. That said, we have further explored their dynamic across metagenomes to contribute to the exchange with the reviewer. For this, we have clustered metagenomes in a similar fashion to their clustering in the manuscript, but this time only using the differential occurrence of invariant genes at the amino acid level (see Supplementary File 2h and 2i). This analysis recapitulated Proteotype D, a set of distantly related metagenomes central to our study.

**Author response image 3. respfig3:** 

As one interpretation, it is possible that the same adaptive genes have passed by recombination across sub lineages in distant geographies (Proteotype D encompasses two sides of Panama as well as Red Sea and north of Indian Ocean). Yet, predominance of the same sub lineage in these regions due to natural selection is also a valid interpretation of the results. As mentioned above, while such analyses may seem to contribute to our study, we believe it would be misleading to suggest invariant proteins used for this figure to have resulted from events of protein sweeps. We hope the reviewer agrees with our resolution.

I found the authors' partitioning of metagenomes into ecologically distinct groups fascinating. And it's particularly interesting that these groups could not be revealed (at least entirely) by analyzing single reads (as shown in Figure 4—figure supplement 3). I'll encourage the authors to venture from their interpretation regarding whole metagenomes to inferring that there appear to be multiple ecologically distinct populations that cause these metagenomes to hold different niches. What might be the ecological distinctions of the populations? Clearly they are different for their temperature adaptation, as revealed by Figure 4 (mislabeled Figure 5). Perhaps the authors could glean something about what is different about the constituent populations from their Figure 4—figure supplement 3, that is by comparing the clusters that they do find with known environmental parameters.

We thank the reviewer very much for their suggestion (and catching the mislabeled Figure 5, which is now fixed). Taking their advice, we have now created a new table (Supplementary File 4d) to investigate and make available to others the ANOVA statistic for each known environmental parameter released by the Tara Oceans Project along with data our study revealed.

As shown in Figure 4, the strong association with latitude can easily be explained by the geographic partitioning of oceanic currents. Besides longitude, temperature best explains clusters of proteotypes as a function of SAAVs. We have updated the results in our revision to cite this table.

Re the concluding statement about "everything is everywhere," I'd temper that by saying 'at least for marine bacteria'. This is because the statement does talk about "everything," so I think there needs to be some limitation to the conclusion.

Fair. The sentence now reads:

“Overall, these findings suggest that environmentally-mediated selection plays a critical role in the journey of cosmopolitan microbial populations in the surface ocean, lending credence to the idea for marine systems that “everything is everywhere but the environment selects” (Baas-Becking, 1934).”

Concerns about evolutionary analyses:In response to point (1), the authors state:"Simply, we observed that certain amino acids (mostly hydrophilic), and most notably, few AASTs (e.g., alanine/isoleucine) predominated in our SAAV table."It has been well-established for decades that hydrophobic amino acids in solvent-inaccessible positions are more conserved (e.g. see reviews cited in the Introduction to Ramsey et al. 2011 Genetics 188(2):478-88). Reproducing this observation in the SAR11 dataset simply confirms that purifying selection can be observed over long timescales. It does not help identify what the changes are that enable ecological adaptation.

Indeed, this section does not contribute to our investigation of ecological adaptation within the SAR11 lineage. It sheds light into purifying selection.

Delmont et al: "These points are interesting, but they do not favor natural selection over neutral evolution. Instead they provide information regarding permissible versus non-permissible diversification"I assume "permissible" refers to purifying selection? Which is an example of natural selection – rather than random, neutral diversification. It is distinct from the adaptive/Darwinian evolution that allows the ecological differentiation of the bacteria. Not separating neutrality, purifying selection and adaptive evolution plagues the whole manuscript. The authors have a strain distribution that appears to reflect ecological differentiation, rather than neutral diffusion. They have a mutation distribution that mainly seems to reflect purifying selection, rather than neutral evolution. Some analyses (e.g. below) suggest adaptive evolution, which is potentially very interesting, but the overall trend is consistent with purifying selection. The authors need to clearly distinguish their different conclusions.

We are extremely thankful for the reviewer for bringing up this point, which made us revise our manuscript carefully to elucidate which aspects of our data are consistent with neutral evolution, purifying selection, and adaptive evolution. We hope that our revisions summarized below will satisfy the reviewer.

First, to convey that a skewed AAST frequency distribution is incompatible with a purely neutral process, but is indicative of purifying selection that is consistent with modern neutral theories of molecular evolution, we have added the following text at the introduction of the AAST frequency distribution:

"While such a skewed AAST frequency distribution cannot be explained by strictly random mutational process (Figure 3B light-gray shaded area), it is compatible with standard theories of neutral or nearly-neutral evolution, since such theories consider the role of purifying selection (Ohta and Gillespie, 1996)."

To support this point and to illustrate how neutral events alone are unable to recapitulate the observed AAST distribution, we have calculated the expected AAST frequency distribution under strictly neutral mutational events by considering the codon frequencies observed in the HIMB83 reference sequence, the edit distance between codons, and the transition/transversion ratio in a probabilistic model (we added a new Materials and methods section "Estimating a neutral AAST frequency distribution" to detail our approach, and updated our reproducible workflow with the details of the procedure). Briefly, we used extremes for the transition probabilities between codons to calculate upper and lower bounds for a strictly neutral AAST frequency distribution, and updated panel B in our Figure 3 to illustrate the expected AAST frequency distribution in the absence of both purifying and adaptive selection (light gray shaded area).

Next, we revised the text to explicitly state and offer interpretation to the apparent role that purifying selection has in maintaining the AAST frequency distribution across distant geographies. We replaced the following text in our previous submission with the following one:

"The overall consistency of AAST frequency distributions across geographies supports the hypothesis that purifying selection maintains amino acid exchangeability within 1a.3.V and enables an interpretation of these data through a neutral model: SAAVs composing the AAST frequency distribution represent primarily neutral mutations that have drifted to measurable levels, and the lack of SAAVs in uncommon AASTs arises due to a strong purifying selection acting against deleterious mutations that occur frequently between dissimilar amino acids."

Moreover, in our previous manuscript we had stated: “Yet, close inspection of AAST frequencies revealed that amino acid exchange abilities subtly diverge in a pattern that correlates with water temperature and/or its co-variables (Figure 3B insets, Figure 3—figure supplement 2).”

But after careful consideration of the reviewer’s point, we found that this statement does not do justice in describing the subtlety of this signal in comparison to the predominant signal of purifying selection. To address this, we have amended the text so that it now reads:

"However, a closer inspection reveals a subtle divergence of amino acid exchangeabilities that correlates with water temperature and/or its co-variables (Figure 3B insets, Figure 3—figure supplement 2). Note that this divergence is AAST specific; for example, positions with mixed proportions of glutamic and aspartic acid are less commonly found in warm waters (linear regression, uncorrected p-value: 2.7×10^(-6)), yet for isoleucine and valine such a correlation is nonexistent (linear regression, uncorrected p-value: 0.418). These findings suggest that amidst a signal that is predominantly indicative of purifying selection, there appears to be a fingerprint of adaptive/divergent processes caused by temperature and/or its co-variables that subtly shift the mutational profile within 1a.3.V."

In addition, in the section entitled “Temperature correlates with amino acid allele frequency trajectories”, we have made more substantial changes that are detailed in our response to the next reviewer concern. Finally, we added a final paragraph that summarizes the ratio between these processes as suggested by the data, which is also described later in this document.

Delmont et al: "we determined that thousands of 1a.3.V allele frequency trajectories correlated with in situ temperature, in line with the biogeography of proteotypes"It is not surprising that many of the variable sites correlate with temperature, because the proteotypes are defined using the variable sites, and the proteotypes correlate with temperature. The authors should seek to synthesize these data – do the variable sites have properties that suggest selection between niches? Do they concentrate in particular genes that might drive adaptation to different niches? Or do they simply correlate with the population structure?

We are again very thankful for this question, which helped us to make a remarkable observation that not only supports our findings but also will be helpful for those who study SAR11 biology. Unfortunately, as of today, scrutinizing all 4,592 positions where amino acid frequencies correlated with temperature with their structural context in protein structures is a computationally intractable problem. However, to tackle this important point we surveyed all genes to quantify the ratio of temperature-correlated and temperature-uncorrelated SAAV positions in each, and then focused on a single gene, which encoded glycine betaine ATP-binding cassette transporter, for an in-depth analysis using its predicted structure. The reason we chose this gene is two-fold. First, betaine transporters of SAR11 are highly translated proteins in the environment that transport osmolyte compounds into cells for energy production, thus playing an essential role in SAR11 biology. Second, this gene exhibited a high ratio of temperature-correlated and temperature-uncorrelated SAAV positions, enabling us to tease apart the distribution of these SAAVs in the structure of the gene while being able to rely on the large number of positions to test whether our observations are statistically meaningful. We laid out our findings in a new paragraph, with this critical section:

“(…) To investigate the positioning of amino acids in the tertiary structure of the permease structure relative to the cellular membrane, we first categorized the location of each residue as transmembrane, cytosolic (inside the inner membrane), or periplasmic (outside the inner membrane) (Figure 3—figure supplement 4). Positions that were not correlated with temperature were commonly transmembrane, and infrequently periplasmic. In contrast, most positions that correlated with temperature were periplasmic (Figure 3—figure supplement 4). The probability of observing a similar distribution between temperature-correlated and temperature-uncorrelated positions across transmembrane, periplasmic, and cytosolic regions was only 0.034 (analytic trinomial test, temperature-uncorrelated distribution as prior), which indicates temperaturecorrelated positions are subjected to unique evolutionary forces. A previous study suggested that periplasmic residues of transmembrane proteins undergo higher rates of adaptive evolution due to their increased exposure to changing environmental conditions (Sojo et al., 2016). This observation lends additional support to the hypothesis that periplasmic SAAV positions within this gene that correlate with temperature are more likely shaped by adaptive processes.”

We also added a new supplementary figure (Figure 3—figure supplement 4) to put this observation into a visual context.

We hope the reviewer appreciates our effort to show that correlation with temperature is not evenly or randomly distributed in the context of this gene. In contrast, there is a significant coupling between the kind of variation and the structural context. We are striving to improve upon current computational limitations to ensure that in the future investigations of such phenomena will be able to explore a wider range of structural diversity.

Overall, I return to my original point from the first round of reviews: is the diversification mainly neutral; dominated by purifying selection; or dominated by adaptive evolution? The most interesting signals may not be from the dominant evolutionary process, of course.

We are certain that the reviewer agrees that quantifying these differential forces accurately is a challenging task. However, we also acknowledge that the reviewer is raising a valid question which will be asked by many readers, and a gross summary is necessary. We addressed this need by including the following paragraph in our revised manuscript to solidify our consideration of the primary evolutionary forces, and to offer a broad summary of our data that shows their ratios. We also used this opportunity to include a limitations statement to warn careful readers with the hope that they will take these suggestions with a grain of salt:

“One question remains: what is the proportion of distinct evolutionary processes acting upon closely related SAR11 populations within 1a.3.V? Offering a precise answer to this critical question is compounded by multiple theoretical and technical factors. […] In summary, this gross summary of the data suggests that among the remarkable amount of variation within some of the most abundant and prevalent microbial populations in the ocean, adaptive evolutionary processes operate on core genes are responsible for variation in about 2% of all codon positions.”

We thank the reviewer for pushing us to better communicate our findings.

In response to point 2, the authors state:Delmont et al: "As far as we know, there is no published study that effectively takes into account multiple SNVs per codon without calculating the exact codon frequencies to estimate synonymity accurately…SNV density was high for 1a.3.V and many SNVs co-occurred in the same codon, rendering classical d(N)/d(S) analyses limited, which had never been challenged with extremely complex environments, if not completely irrelevant."Yang (2007) MBE 24(8):1586-91 – with over 6,000 citations – describes how baseml can be used to calculate d(N)/d(S) using base substitution, rather than codon substitution, models. This accounts for multiple substitutions per codon, and provides a statistical test for evidence of selection versus neutrality, unlike the methods presented in the current version. It does not matter how complex the environments are – indeed, d(N)/d(S) has been applied to the question of SAR11 adaptation to different ecologies by Brown et al., 2012 and Luo and Hughes, 2012, Mol. Syst. Biol. 8:625. The authors should at least apply standard methods to the assembled genomes, based on the ecological separation they have determined. Methods also exist for calculation of this statistic from metagenomic data, and might be more informative.

We apologize for the vague statement. Indeed, when performing comparative genomics, it is possible to calculate d(N)/d(S) even in the case of multiple substitutions per codon. The critical point here is that both Yang as well as Brown et al. relied on comparative genomics to study genes under positive selection, i.e., they did have genomes to compare with each other (for instance, Brown et al. wrote in their manuscript: “Signatures of positive selection were detected in the two polar genomes compared with their most closely related tropical counterparts”). Our study differs from these studies and other previous attempts as it deals with novel challenges that arise when working with metagenomes. To address these additional challenges that none of the previous studies addressed, we developed a new workflow to determine SAAVs regardless of the number of SNVs per codon in a given metagenome without relying on comparative genomics. We regret that our previous responses may have compounded this misunderstanding; however, the manuscript clearly states that our contribution is from the perspective of metagenomics rather than comparative genomics:

“SAAVs: Accurate characterization of non-synonymous variation

Percent identity distributions are useful to assess overall alignment statistics of short reads to a reference; however, they do not convey information regarding allele frequencies, their functional significance, or association with biogeography. […] Thus, quantifying frequencies of full codon sequences is a requirement to correctly assess synonymity, and it is this principle we employed in our workflow to extract SAAVs.”

We hope this clarification will satisfy the reviewer.

Delmont et al: "The significant shift in SAAVs/SNVs ratio between cold currents and warm currents is of relevance to our study, especially in light of other insights (especially, the linkage between allele frequency trajectories and in situ temperature, and the biogeography of proteotypes), favors the predominant role of natural selection acting on the permissible amino acid diversification traits of this global SAR11 lineage."The correlation between allele frequencies and temperature, and proteotype and temperature, is confounded by the correlation between proteotypes and allele frequencies. This should be explained (if this is wrong, please also make that clear in the text).

We agree with the reviewer that allele frequency trajectories confound proteotype classification, and so we have removed the sentence, “This is corroborated by thousands of allele frequency trajectories correlated with in situ temperatures” from the text.

The authors need to make clear whether the natural selection they are referring to is stronger/weaker purifying selection changing the SAAV/SNV ratio; or positive selection driving adaptive diversification increasing the SAAV/SNV in some populations.

We agree that the manuscript needed to clearly distinguish between varying levels of purifying selection versus adaptive evolution, and we hope the reviewer is satisfied with our consistency in delineating the evolutionary forces that can be gleaned through our data in the latest version of our manuscript. In the specific case of the SAAV/SNV ratio, we have opted to remove our SAAV/SNV ratio analysis from our revision altogether. Our rationale stems from planned future efforts to create a more robust and meaningful metric, in contrast to the current one that has weaknesses as the reviewer pointed. Nevertheless SAAV/SNV ratio was only ever briefly mentioned in our study, and the exclusion of it does not influence our findings. We thank the reviewer for pointing this.

Delmont et al: "Regarding the fixation index, we agree with the reviewer that this would have been a valuable addition to our work…We believe the significant amount of methodological development in this work that offers descriptions of the biogeography of the most abundant lineage in surface oceans conveys a story that will not benefit from the inclusion of additional analyses. "I don't understand how this statistic can be "a valuable addition", yet the story "will not benefit from the inclusion of additional analyses.". The deep learning analysis is still not intuitively explained in the new version (not the authors' fault, as these methods are designed to be complex). It should be complemented by fixation index calculations, which are simple, and would allow the readers a much greater insight into the basis of the results.

We thank the reviewer for their persistence in benchmarking the Deep Learning method against classically used and well-understood methods. We have now calculated fixation indices for the SAAV data, and created a new computational tool to make sure this approach is accessible to other researchers in a more straightforward manner, as well. The results recapitulate our primary observations of distant geographies being linked, and we have added a new supplementary figure (Figure 4—figure supplement 4) comparing the dendrograms and placement of the clusters onto the world map.